# The origin, evolution, and molecular diversity of the chemokine system

Alessandra Aleotti[1,2,*] ⓘ, Matthew Goulty[1,2,*], Clifton Lewis[1,2] ⓘ, Flaviano Giorgini[1,2], Roberto Feuda[1,2] ⓘ

Chemokine signalling performs key functions in cell migration via chemoattraction, such as attracting leukocytes to the site of infection during host defence. The system consists of a ligand, the chemokine, usually secreted outside the cell, and a chemokine receptor on the surface of a target cell that recognises the ligand. Several non-canonical components interact with the system. These include a variety of molecules that usually share some degree of sequence similarity with canonical components and, in some cases, are known to bind to canonical components and/or to modulate cell migration. Whereas canonical components have been described in vertebrate lineages, the distribution of the noncanonical components is less clear. Uncertainty over the relationships between canonical and noncanonical components hampers our understanding of the evolution of the system. We used phylogenetic methods, including gene–tree to species–tree reconciliation, to untangle the relationships between canonical and noncanonical components, identify gene duplication events, and clarify the origin of the system. We found that unrelated ligand groups independently evolved chemokine-like functions. We found noncanonical ligands outside vertebrates, such as TAFA "chemokines" found in urochordates. In contrast, all receptor groups are vertebrate-specific and all—except ACKR1—originated from a common ancestor in early vertebrates. Both ligand and receptor copy numbers expanded through gene duplication events at the base of jawed vertebrates, with subsequent waves of innovation occurring in bony fish and mammals.

## Introduction

The chemokine system is responsible for regulating many biological processes, including host defence, neuronal communication, and homeostasis (1, 2, 3, 4, 5). The system has two components, a ligand, usually a small cytokine called a chemokine, and a receptor. It typically operates through chemoattraction, wherein one cell type produces and secretes chemokines, creating a chemical gradient as these molecules disperse. Cells equipped with the corresponding chemokine receptors on their membranes can recognise and bind to specific chemokines, promoting their migration along the gradient (4). This mechanism allows cells to reach target locations, such as infection sites during inflammation or tissues important for homeostatic functions, for example, leukocyte maturation and trafficking (3, 6). Chemokines involved in the latter homeostatic functions are usually constitutively expressed, whereas those involved in inflammatory responses have an inducible expression (7). Chemokine ligands are categorised into four groups, XC, CC, CXC, and CX3C, according to the pattern of cysteine residues in the N-terminal portion of the protein (8). Likewise, the receptors are classified based on the ligands they bind to into four groups, the XCR, CCR, CXCR, and CX3CR, and all of them belong to the GPCR class A superfamily (9). In addition to canonical components, other molecules have been discovered to function similarly to chemokine ligands (1) or receptors (2) (see Table 1). These include the following: the chemokine-like factor (CKLF) that binds to chemokine receptor CCR4 (10, 11) and drives cell migration in vivo (12); TAFA chemokines, expressed mainly in the nervous system, which share structural similarities to canonical chemokines (25, 26) and bind GPCRs related to chemokine receptors, for example, formyl peptide receptors (FPR) (27, 28) and GPR1 (29); cytokine-like 1 (CYTL1) that binds CCR2 (22) and has been suggested to be related to CC ligands based on the presence of a IL8-like chemokine fold (40). There are also noncanonical chemokine receptors, such as the following: the chemokine-like receptor (CML1, or also CMKLR1) (36); atypical chemokine receptors (ACKRs) (33); and viral chemokine receptors (41, 42, 43, 44). Unlike other chemokine receptors, atypical receptors cannot initiate classical chemokine signaling upon ligand binding (33, 45). The human genome encodes four types of ACKRs: the ACKR1 (also known as DARC), ACKR2 (also known as D6), ACKR3 (also known as CXCR7), and ACKR4 (also known as CCRL1) (34, 35). In addition, several proteins of viral origins, such as US28 from human cytomegalovirus, have chemokine-receptor/binding activity (41, 42). These viral proteins can bind a wide array of chemokine ligands (42).

Despite the extensive research on the chemokine system, with over 320,000 articles available on PubMed, many aspects of its evolution remain unclear. For instance, the homology between canonical and noncanonical ligands is uncertain and supported by circumstantial evidence, such as shared specific motifs (12, 25, 40,

[1]Neurogenetics Group, University of Leicester, Leicester, UK   [2]Department of Genetics and Genome Biology, University of Leicester, Leicester, UK

Correspondence: aa1176@leicester.ac.uk; mg478@leicester.ac.uk; rf190@leicester.ac.uk
*Alessandra Aleotti and Matthew Goulty contributed equally to this work

**Table 1.** Summary table of all the canonical and noncanonical chemokine components analyzed in this study.

| | Names | Abbreviations | *H. sapiens* orthologs | Functions | References |
|---|---|---|---|---|---|
| **Ligand Groups** | Canonical chemokines | CCL, CXCL, XCL, CX3CL | CCL1-3, 3L1, 3L3, 4, 4L1-L2, 5, 7, 8, 11, 13, 14-28; CXCL1-4, 4L1, 5-14, 16,17; XCL1,2; CX3CL1 | - Chemokine receptor binding and signalling<br>- Chemoattraction of leukocytes<br>- Homeostasis of leukocytes | (2, 4, 7) |
| | CKLF-like MARVEL transmembrane domain-containing proteins (chemokine-like factor super family) | CKLF, CMTM | CKLF1; CMTM1-8 (CKLF, CKLFSF1-8) | - CKLF1 (CKLF) binds to chemokine receptor CCR4<br>- CKLF1 (CKLF): chemotactic activity for lymphocytes, macrophages, and neutrophils<br>- Other CMTMs: variably expressed in immune system; putative roles in immunity, programmed cell death, regulation of antitumour immunity etc. | (1, 10, 11, 12, 13, 14, 15, 16, 17, 18, 19, 20, 21) |
| | Cytokine-like protein 1 (Protein C17 or C4orf4) | CYTL | CYTL1 | - Chemokine receptor binding (CCR2) and signalling<br>- Chemoattraction monocytes/macrophages<br>- Chemotactic activity in neutrophils | (1, 22, 23, 24) |
| | TAFA chemokines (family with sequence similarity 19 (chemokine (C-C motif)-like) member A) | TAFA | TAFA1-5 (FAM19A1-5) | - Formyl-peptide receptor binding and signalling (TAFA4 and 5)<br>- Putative binding to other GPCRs: GPR1 (TAFA1); S1PR2 (TAFA5)<br>- Expressed in central and peripheral nervous system<br>- Implicated in vast diversity of physiological processes | (1, 25, 26, 27, 28, 29, 30, 31, 32) |
| **Receptor Groups** | Canonical chemokine receptors | CCR, CXCR, XCR, CX3CR | CCR1-10; CXCR1-6; XCR1; CX3CR1 | - Chemokine binding and signalling<br>- Chemotaxis of leukocytes<br>- Homeostasis of leukocytes | (2, 4, 7) |
| | Atypical chemokine receptors | ACKR | ACKR1-4 (DARC; D6; CXCR7; CCRL1) | - Chemokine binding, but no signalling<br>- Resolution of inflammatory response | (33, 34, 35) |
| | Chemokine receptor-like (chemokine (C-C motif) receptor-like 2) | CCRL | CCRL2 (ACKR5) | - Binds CCL5 and CCL19, but no signalling<br>- Binds chemerin and presents it to CMKLR1 | (36, 37) |
| | Chemokine-like receptor 1 | CML | CML1 (CMKLR1; ChemR23) | - Binds chemerin inducing migration of macrophages and dendritic cells<br>- Binds also other anti-inflammatory molecules (e.g., Resolvin E1 (RvE1)) | (36) |
| | Formyl-peptide receptors | FPR | FPR 1-3 | - TAFA chemokine binding<br>- Chemoattraction, modulation of inflammation | (27, 28, 38) |
| | Putative chemokine receptors | ACKR6, CXCR8 | PTITMP3, CXCR8 (GPR35) | - ACKR6/PTITMP3 binds CCL18 (NB: It is not a GPCR)<br>- CXCR8/GPR35 binds CXCL17 | (37, 39) |

46). Furthermore, the relationships between canonical, atypical, and viral receptors and the outgroup of the canonical chemokine receptors remain uncertain. Finally, the evolutionary history of the canonical and noncanonical components remains poorly understood outside a few key model systems (9, 47, 48). These outstanding questions share common underlying causes, including the use of inadequate inference methods (such as relying solely on sequence similarities) and limited sampling of species (e.g., focusing mainly on humans, mice, and zebrafish (7, 49)). In addition, solving the phylogenetic relationships for short molecules such as chemokine receptors and ligands is particularly challenging because of the lack of strong phylogenetic signals (50).

Here, to clarify these outstanding questions, we use state-of-the-art phylogenetic methods, including those designed for single-gene phylogenies, a large taxonomical sampling comprising both vertebrate and invertebrate genomes and the entire complement of canonical and noncanonical components of both receptors and ligands. Our findings substantially clarify the phylogenetic relationship between canonical and noncanonical ligands and receptors and suggest that unrelated proteins evolved "chemokine-like" ligand function multiple times independently. In addition, we discovered that all the canonical and noncanonical chemokine receptors (except ACKR1) originated from a single duplication in the vertebrate stem group, which also gave rise to many GPCRs. Finally, we characterized the complement of canonical and noncanonical components in the common ancestor of vertebrates and identified several other ligands and receptors with potential chemokine-related properties that could be explored in future functional work.

## Results

### There are five unrelated groups of ligands

Initially, we focused on the ligands, including all the canonical chemokines, the CYTL, the TAFAs, and the CKLF Super Family (CKLFSF) proteins (Table 1). The presence of a four-transmembrane MARVEL domain in the latter proteins (12, 13, 14) distinguishes them from canonical chemokines, the CYTL and the TAFAs. Therefore, we separated these two groups for further analysis. Using BLASTP or PSI-BLAST (51, 52, 53) (see the Materials and Methods section for more details) against 64 species from 19 animal phyla (Table S1), we identified 891 putative homologs for chemokines, TAFA, and CYTL and 602 putative homologs of the CKLF Super Family.

We used Cluster Analysis of Sequences (CLANS) (54, 55), a clustering tool based on sequence similarity and local alignment, to identify homology within these two groups. Unlike traditional phylogenetic methods, CLANS assigns homology between sequences based on BLAST and customisable stringency levels defined according to P-values (54). When two (or more) sequences are connected at a lower P-value (closer to 0), this indicates a high level of homology. Conversely, if two or more sequences only connect at a higher P-value, this suggests a relatively low level of sequence homology. Our analysis shows that canonical chemokines form a distinct group with a clear distinction between C-X-C-type and C-C-type (Fig 1A), whereas CXCL17, TAFA, and CYTL remain separate from

canonical chemokines and from each other even at the loosest P-values tested (Fig 1A). The distinction between CXCL17 and all other canonical chemokines is consistent with our receptor results, showing that the potential receptor for CXCL17, GPR35 (39), is also not within the canonical chemokine receptor group (see below). However, it is important to note that recent studies fail to demonstrate CXCL17 activity at GPR35 (56, 57). Within the CKLFSF, two large clusters were identified, named CKLF I and CKLF II, although these ultimately connect to form one large superfamily (Fig 1B). These clusters are robust to the different stringency thresholds used (Figs S1 and S2 and see the Materials and Methods section for further details). Our results indicate that even when the stringency level to detect homology is relaxed, canonical chemokines, TAFA, CYTL, and CXCL17 remain in distinct clusters. This suggests that, similarly to CKLFs, these proteins are not homologous and convergently evolved chemokine-like properties. We have thus identified five distinct groups of ligands: (i) the canonical chemokines, (ii) TAFA "chemokines," (iii) CYTL, (iv) CXCL17, and (v) CKLF Super Family (Fig 1A and B).

### The evolution of chemokine and chemokine-like ligands in animals

To better understand the evolution of both canonical and noncanonical chemokine ligands, we performed a separate phylogenetic reconstruction for each group (Figs 1C and D and S3, S4, S5, S6, S7, S8, S9, S10, S11, and S12) (see the Materials and Methods section for details). To evaluate the nodal support, in addition to the UltraFast bootstrap (UFB) (58, 59), we used Transfer Bootstrap Expectation (TBE), a method that has been developed for single-gene phylogeny (60). To evaluate ortholog/paralog relationships and overall dynamics of the ligand complement, we used GeneRax (61). This method uses maximum likelihood to reconcile the gene tree with the species tree (61). In brief, given a gene and species tree, GeneRax uses a maximum likelihood approach to optimise the duplication and loss events (61, 62 Preprint).

Our analysis initially identified a few invertebrate putative chemokine ligands (Fig 1A), however, these sequences lacked protein signatures associated with the canonical ligands (Figs S13, S14, and S15 and Supplementary File 3 in the GitHub repository: Roberto-Feuda-Lab/Chemokine2023 (github.com)), and they were therefore excluded from further analysis (see the Supplementary results section for further information). The phylogenetic tree for the canonical ligands identifies two major groups, the CC-type, which also includes the XC and X3C types, and the CXC type (TBE = 0.95, UFB = 92%) (Figs 1C and S3 and S4), confirming the previous finding obtained using synteny data (63, 64). Next, to clarify the distribution of canonical chemokines, we first reconciled their gene tree with the species tree and then used the reconciled tree to trace the presence/absence of each chemokine group throughout all the species (Figs 2A and S16). Our results confirm previous findings that canonical chemokines are uniquely present in vertebrates (47, 63). In addition, they indicate that chemokines are not evenly distributed across vertebrates and can be different even between closely related species (65). Some are very ancient, for example, CXCL12 is present in lamprey; CXCL14 and CCL20 are present in all jawed vertebrates; and CXCL8 is present throughout bony fishes

and tetrapods, with few exceptions, notably mice and rats. However, a large part of the chemokine diversity evolved within mammals (e.g., CXCL1/2/3, CXCL16, and CCL25), particularly placentals (e.g., CXCL5/6 and CCL3/18). The phylogenetic relationships we uncovered in our reconciled tree were mostly compatible with known syntenic relationships as described in human (7). For example, the large cluster of *CXC*-type chemokine genes present in human chromosome 4 contains *CXCL1-11* plus *CXCL13* (7), all of which coalesce within a monophyletic group in our tree (Fig 2A). The microsynteny within this cluster is also, to some extent, reflected in the phylogenetic relationships. Similarly, the other large syntenic cluster of chemokines, located on human chromosome 17, containing most of the *CC*-type chemokines (7), corresponds, with few exceptions, to a large monophyletic clade in our tree (Fig 2A). *CXCL16* which is on a nearby locus of chromosome 17, is also phylogenetically related to this *CC*-type clade (Fig 2A). The complement of the canonical chemokines undergoes the largest expansion at the base of jawed vertebrates, where there is an expansion from 4 to 18 genes (Fig 2B). A second expansion occurred at the base of bony fishes (i.e., Osteichthyes) followed by relative stability until placental mammals, where the total number of canonical chemokine ligands jumped to 45 genes. Finally, unlike previous works (66), our results support the presence of orthologs of both CC type and CXC type in the common ancestor of all vertebrates (Fig 2A).

Differently from the canonical chemokines, we identified a bona fide TAFA, that is, with specific protein motifs, in the urochordates, the sister group to vertebrates (see the Supplementary results section and Figs S17 and S18). The phylogenetic trees (Figs S5 and S6) identified monophyletic groups for TAFA5 (TBE = 0.98, UFB = 98%), TAFA1 (TBE = 0.94, UFB = 98%), TAFA4 (TBE = 0.77, UFB = 75%), and TAFA2/3 (TBE = 0.65, UFB = 84%). The reconciled tree from GeneRax places the root at the urochordate sequence (Fig S19), therefore clarifying that the TAFA5 clade is the sister group to TAFA1-4 (Fig 2A). The family originated in the ancestor of urochordates and vertebrates, and the first duplications occurred at the base of vertebrates giving rise to the TAFA5 split followed by the TAFA1 split. Subsequently, at the base of jawed vertebrates, additional duplications bring the complements from 3 to 10 (Fig 2B), giving rise to the remaining groups so that all jawed vertebrates possess the full diversity of TAFAs.

The phylogenetic trees for CYTL and CXCL17 mainly reflect the species trees (Figs S7, S8, S9, and S10), and the reconciliations revealed very simple complement dynamics (Figs 2B and S20 and S21). However, these molecules show a remarkable difference in their distribution. CYTLs are present throughout gnathostomes, whereas CXCL17 is found only in placental mammals (Fig 2A).

The phylogenetic analysis for the CKLF super family (Figs 1D and S11 and S12) recovered a monophyletic clade for the CKLF I group (TBE = 0.96, UFB = 80%) that we had already identified through CLANS. This group contains CKLF, which is known to interact with C-C chemokine receptor 4 (10, 11), and CMTM1, 2, 3, 5, and proteolipid protein 2 (PLP2). Other monophyletic clades that are consistent with the CLANS are CMTM4/6 (TBE = 0.90, UFB = 61%), CMTM7 (TBE = 0.92, UFB = 83%) and a clade containing CMTM8 plus other related molecules such as plasmolipin (PLLP) and myelin and lymphocyte proteins (MAL) (TBE = 0.89, UFB = 60%). The latter were all part of a large cluster that we called CKLF II in the CLANS (Fig 1B). However, the placement of the root of the tree in Fig 1D can affect the interpretation of the

relationships among CKLF II subgroups. To address this problem and clarify the patterns of duplications and the presence/absence of each group throughout animals, we used GeneRax to reconcile the gene with the species tree (see above and Material and Methods section for details). Our results suggest (Figs 2 and S22) that most CKLFSF groups, such as CMTM4, 6, and 8, originate in the vertebrate stem group from preexisting CMTM genes and are widely distributed in animals. The CKLF I subgroups originate from duplications at the base of jawed vertebrates, except for the split between CKLF and CMTM1 that occurs only within mammals (Fig 2A). We observe the major two expansions of the CKLFSF genes in the stem group of vertebrates (from 6 to 10 complements), and then in jawed vertebrates (from 10 to 16 complements). Interestingly, the extent of these expansions is less drastic than those we see for canonical chemokines (Fig 2B). In total, we have identified that the five distinct ligand groups have a different origin in the animal tree of life and underwent divergent evolutionary histories.

## Canonical and noncanonical chemokine receptors are divided into four groups

Next, we investigated the origin and pattern of duplication for the chemokine receptors and chemokine-like receptors (Table 1). Using BLASTP against the 64 species, we identified 7,157 putative chemokine receptors (see the Materials and Methods section for more details) and investigated their relationships using CLANS (see above for justification). The result (Fig S23C) identifies four main groups of chemokine receptors and chemokine-like receptors. The first comprises canonical receptors (i.e., CCR, CXCR, CX3CR1, CX3C, and XCR1), and the second includes atypical receptor 3 and GPR182, which has been recently shown to have chemokine receptor activity (67). The third group, which we named Chemokine-like plus (CMLplus), contains the chemokine-like receptors (CML1 also known as chemerin receptor 1), FPR that bind the TAFA ligands (27, 28) and other GPCRs such as GPR1 (chemerin receptor 2), GPR33, PTGDR2. Furthermore, the CLANS analysis identifies an intermediate group containing angiotensin, apelin, and other receptors and shows sequence similarity to canonical and chemokine-like receptors (Fig S23B). Finally, our analysis identifies a small cluster composed of only ACKR1 that do not connect to other GPCRs or other atypical receptors even at loose *P*-value thresholds. This indicates that their sequence is either nonhomologous or highly divergent from other chemokine receptors and atypical receptors. Overall, these groups are robust to the stringency threshold used (i.e., different *P*-values) (Fig S23A–C). Interestingly, no specific cluster of viral or viral-like receptors was identified, but six of the reference viral receptor sequences clustered with the canonical chemokine receptors.

Altogether, these results confirm the homology between the canonical receptors and atypical receptor 3/GPR182. However, these findings indicate that the other GPCRs, such as the chemokine-like receptors, FPR, GPR1, and GPR33, are also closely related to the canonical receptors. Remarkably, these results also indicate that ACKR1 is not homologous to the canonical chemokine receptors. Furthermore, all clusters of chemokine receptors contained only vertebrate sequences, except for the receptors of viral origin.

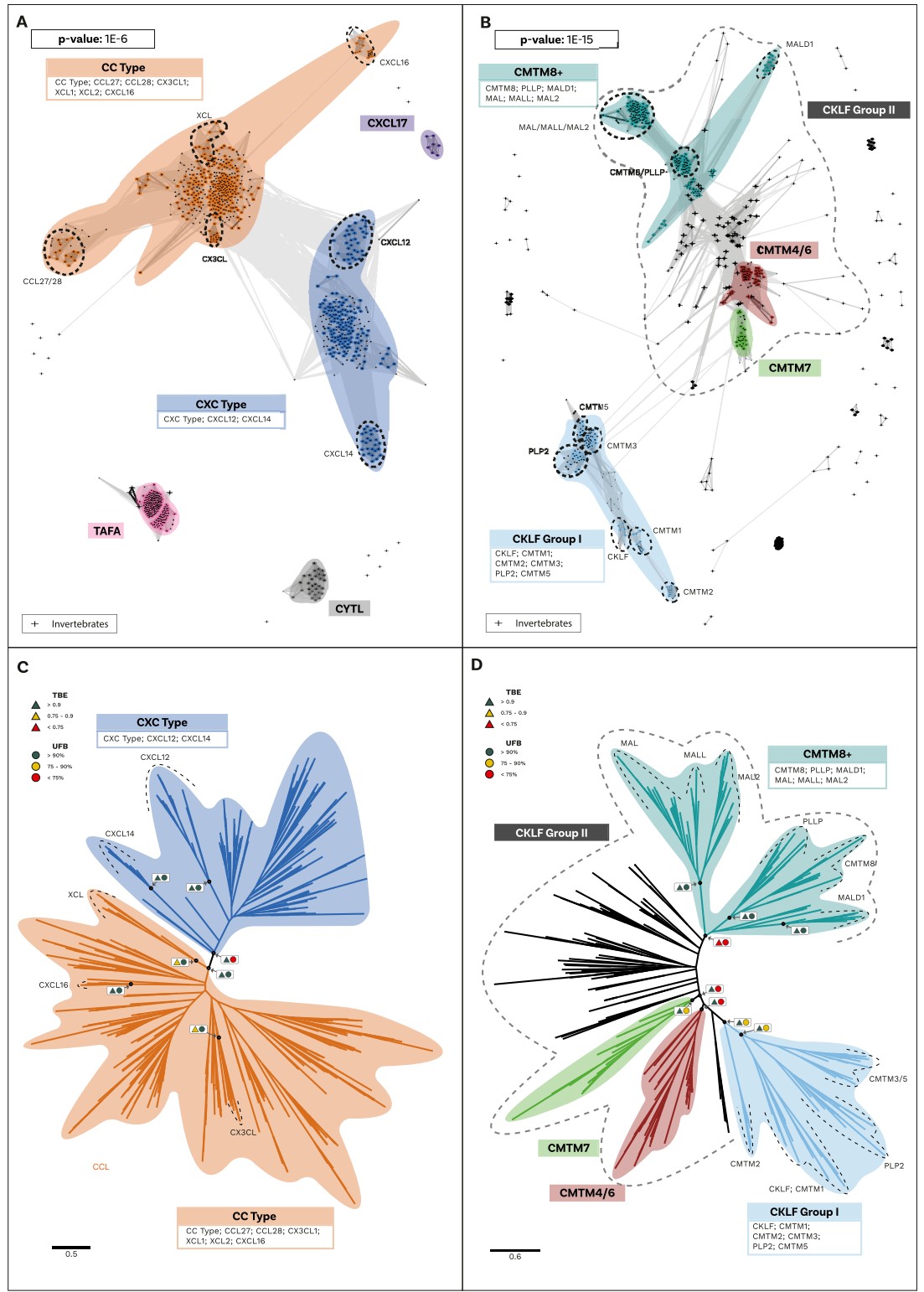

**Figure 1. Cluster Analysis and phylogeny of ligand groups.**
**(A)** Similarity-based clustering, using Cluster Analysis of Sequences, of canonical chemokines and related molecules with sequence similarity. Canonical chemokines are an independent group from other related molecules (TAFA, CYTL, and CXCL17). Canonical chemokines are composed of two large groups (CC type and CXC type) within which some divergent subgroups are highlighted. The clustering and connections shown are at the $P$-value threshold of $1 \times 10^{-6}$. Other $P$-values tested are shown in Fig S1. Candidate invertebrate sequences are shown as crosses and further information regarding them can be found in the Supplementary results section. **(B)** Similarity-based clustering, using Cluster Analysis of Sequences, of the chemokine-like factor (CKLF) super family (CKLFSF). Two major clusters are formed: the smaller "CKLF Group I"

## Canonical and chemokine-like receptors derive from single-gene duplication in the ancestor of vertebrates

Previous studies suggested that the chemokine receptors evolved from a duplication of angiotensin receptors (68) or adrenomedullin receptors (47, 69). However, these works were based on error-prone phylogenetic methods such as Neighbour Joining (69). Our CLANS results indicate that chemokine receptors and chemokine-like receptors have only been observed in vertebrates. Therefore, we focused on invertebrate genomes to clarify the chemokine receptor's outgroup. To clarify this, we lowered the *P*-value thresholds of CLANS (to *P*-value < 1 × 10$^{-50}$) and collected a combined dataset including all chemokine receptor sequences and outgroups (i.e., sequences that connect to the chemokine receptor cluster), resulting in 3,026 sequences. We then performed a phylogenetic tree on this dataset using maximum likelihood methods with UFB and TBE for evaluating nodal support (see above and Materials and Methods section for details).

Our combined phylogenetic analysis shows strong support for the monophyly of canonical chemokine receptors (UFB = 96, TBE = 1.0), the CML-plus (UFB = 95, TBE = 0.99) and the atypical 3/GPR182 (UFB = 100, TBE = 1) (Figs 3 and S24 and S25). In contrast, viral chemokine receptors are paraphyletic, with three sequences placed within the canonical chemokine receptors and three forming a monophyletic group sister to them (UBF = 84 TBE = 1.0). Our results also suggest that the intermediate group, which includes apelin receptors, angiotensin receptors, bradykinin receptors, and orphan GPCRs (e.g., GPR25; GPR15) forms a monophyletic clade with the canonical chemokine receptors, CML-plus group and atypical3/GPR182 (UFB = 61, TBE = 0.91). However, its position changes between the sister group to canonical chemokine receptors plus atypical3/GPR182 in the TBE tree (TBE = 0.84) and sister to CML plus in the UFB tree (UFB = 38).

All the groups mentioned above form a large clade composed of vertebrate-specific GPCRs (UBF = 100 TBE = 0.96) that also includes other GPCRs, such as CLTR and P2RY receptors (Figs 3 and S24 and S25). Another orphan GPCR, GPR35, had been proposed as a potential chemokine receptor (39); however, this was later questioned (56, 57) and GPR35 is still generally considered orphan (70, 71, 72). Our analysis collected GPR35 and placed it within this large vertebrate-specific clade indicating that is it also a vertebrate-specific gene but not phylogenetically a "canonical" chemokine receptor. The closest outgroup to this clade is composed of a few sequences from urochordates, the sister group of vertebrates (UFB = 49 TBE = 0.91) (Figs 3 and S24 and S25). Interestingly, as the sister group of this clade, we identify a group composed of Relaxin receptors, which contain sequences from both urochordates and vertebrates (UBF = 53 TBE = 0.95). Finally, as the sister group of these large clades, we identified a clade of cephalochordate-specific sequences (UBF = 44).

To clarify the duplication pattern and origin of the chemokine receptors, we used GeneRax (61) (see the Materials and Methods section). Our results indicate (Figs 4A and S26) that all chemokine receptors (except ACKR1) originated from a duplication in the stem lineage of vertebrates. This duplication of an unknown GPCR gave rise to the CML-plus, the canonical chemokine receptors atypical 3/GPR182 groups and the intermediate group and other GPCRs (Figs 4A and S26). This result is consistent with the distribution of the paralogous Relaxin receptors which are present both in urochordates and vertebrates and the position of the orphan urochordate sequences as the sister group of canonical chemokine receptors, CML-plus group, and atypical3/GPR182 and other GPCRs (see above). Furthermore, the phylogenetic relationships among canonical chemokine receptors are overall consistent with the syntenic gene patterns known in human (7). The largest cluster of chemokine receptor genes spans 3 closely located loci on human chromosome 3 (7). It includes most *CCRs*, *XCR*, and *CX3CR* and corresponds to one of the two major monophyletic clades in our tree (Fig 4A). Another example is the mini cluster of *CXCR1* and *CXCR2*, located on human chromosome 2 (7), which we also found to form a monophyletic clade (Fig 4A).

We used the reconciliation to better understand the repertoires of receptors present at key nodes during vertebrate evolution. Our results (Fig 4B) show a substantial difference in the duplication pattern of different receptor families. For example, the complement of the atypical3/GPR182 remains constant throughout vertebrate evolution, whereas the canonical and chemokine-like receptor groups expanded dramatically. The canonical chemokine receptors expanded from 5 to 20 genes and the CML-plus from 1 to 11 in the ancestor of the jawed vertebrates (Fig 4B). The expansion of the canonical CKRs is also not evenly distributed across its subgroups, with the ancestral CC type receptors undergoing a series of duplications in jawed vertebrates, whereas the CXCR paralogs did not, specifically one (CXCR4) remains in single copy across all vertebrates. We inferred that in the stem lineage of vertebrates, five canonical chemokine receptor paralogs had already diverged, representing the two major types of receptors (2 CCR and 3 CXCR paralogs). Also present in the stem lineage of vertebrates were ACKR3 and GPR182 and a single-copy gene, which would later diverge to produce all the CML-plus clade.

## Discussion

This work substantially clarifies the evolutionary assembly of the chemokine system. Our analysis shows that contrary to the receptors which evolved from a single duplication event in the vertebrate stem group, several unrelated molecules acquired the

---

and the heterogenous "CKLF group II" that also includes some invertebrate sequences (shown as crosses). Subclades, including the known members of the CKLF super family, are highlighted. The clustering and connections shown are at the *P*-value threshold of 1 × 10$^{-15}$, as this is the threshold at which the two major clusters connect. Other *P*-values tested are shown in Fig S2. **(C)** Maximum-Likelihood un-rooted phylogenetic tree of canonical chemokines. CC type and CXC type are split into two separate clades. Supports for key nodes are indicated in boxes with Transfer Bootstrap Expectation represented by triangles and the Ultrafast Bootstraps as circles. A traffic light colour code is used to indicate the level of support: high (green); intermediate (yellow), and low (red). **(D)** Maximum-Likelihood un-rooted phylogenetic tree of the CKLF super family (CKLFSF). The CKLF group I is monophyletic, whereas the CKLF group II is not. Supports for key nodes are indicated in boxes with Transfer Bootstrap Expectation represented by triangles and the Ultrafast Bootstraps as circles. A traffic light colour code is used to indicate the level of support: high (green), intermediate (yellow), and low (red).

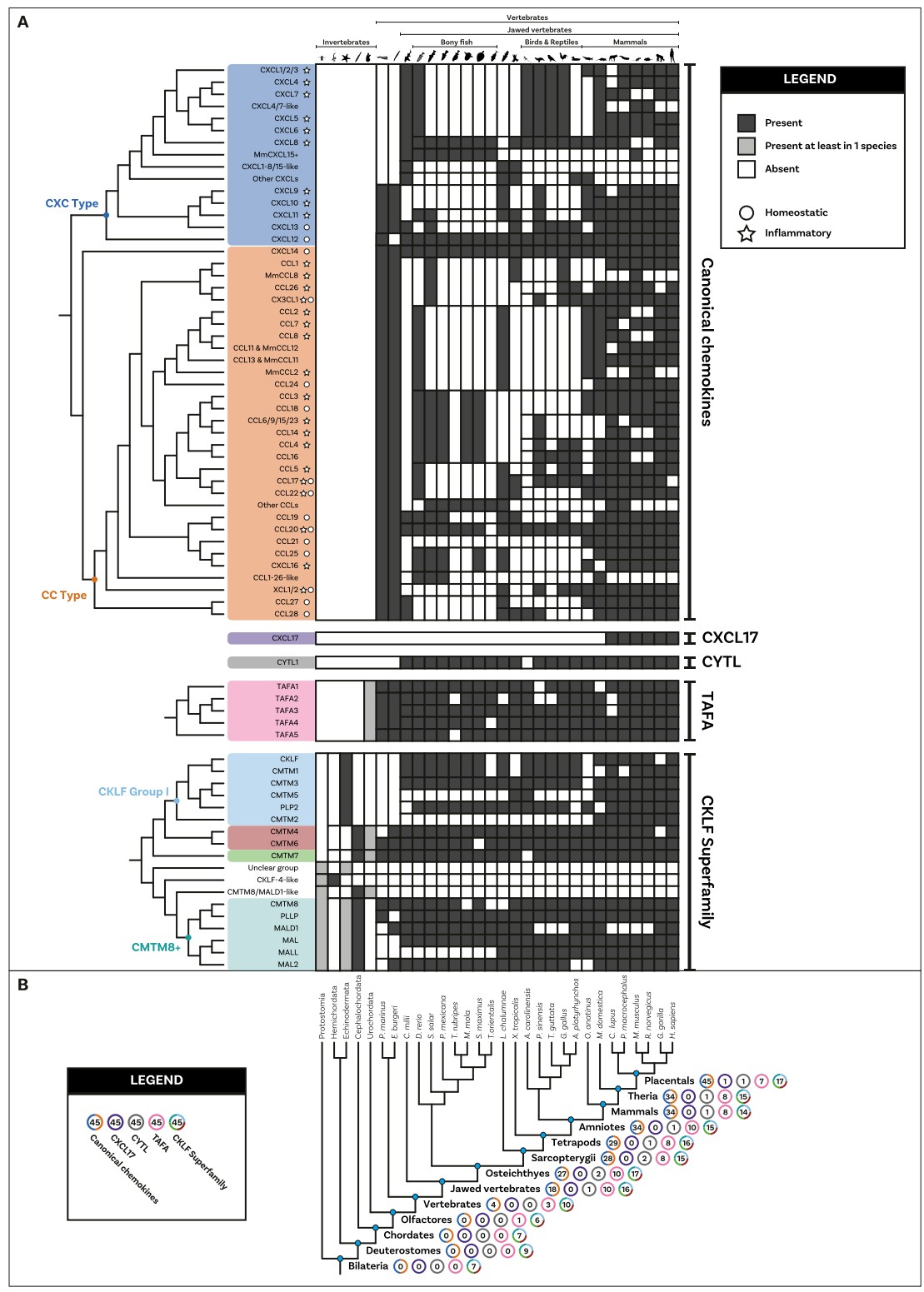

**Figure 2. Distribution and duplication patterns of ligand groups.**
**(A)** Presence of all ligand groups are mapped onto a species tree. Gene trees and duplication events are based on the gene tree to species tree reconciliation analyses. The nomenclature for canonical chemokines is primarily based on known chemokines of human (or mouse). Where human and mouse chemokines do not correspond, the default name refers to the human gene and the mouse (*Mus musculus*) one is indicated with "Mm." Chemokines that have been classically described as having either homeostatic or inflammatory function are indicated with a circle or a star, respectively. The classification used here was based on reference 7 with the inflammatory type also including chemokines they described as plasma/platelet types. Overall, canonical chemokines originated in vertebrates and expanded a first time in jawed

ability to interact with chemokine receptors over the course of evolutionary history. Furthermore, our results (summarized in Fig 5) suggest that the key components of the chemokine system, including the chemokine receptors themselves, evolved in the stem group of vertebrates in the Cambrian around 500 million years ago and then underwent substantial diversification in the stem group of jawed vertebrates. These findings shed new light on the complex evolutionary history of the chemokine system.

### Unrelated molecules converged to chemokine function

Based on the presence of shared protein motifs, TAFA "chemokines" (25, 26), CXCL17 (76, 77) and CYTL (40) have been proposed to be homologous to chemokine ligands. However, our findings strongly suggest that these molecules are not homologous (Fig 1) and likely acquired the ability to activate a chemokine-like response through convergent evolution. Our conclusions differ from those previous studies (25, 40, 76, 77) because of the differences in data completeness and methodological approach. Specifically, we used a complete set of canonical and noncanonical ligands and assessed the homology using overall sequence similarity rather than single motifs. Our results support and expand upon the findings of (46), which suggested that the presence of a CXC or CC motif is necessary but not sufficient for a protein to be defined as a chemokine ligand. Similarly, CKLF has been considered a "new member" of the chemokine family based on its function (12); we argue that classification based solely on function is insufficient and can be misleading. Instead, we recommend considering the evolutionary relationships among these molecules as the primary criterion for classification.

### Most of the canonical and noncanonical ligands are vertebrate innovations

Our results clarify the distribution of canonical chemokine ligands in animals (Fig 2) and confirm that they are present only in vertebrates (47). We identify orthologs of CXCL and CCL ligands in both extant lineages of cyclostomes (Fig 2A). Although chemokines have already been described in lamprey (66, 78, 79), it is the first time, to the best of our knowledge, that they are described also in hagfish. Our findings also indicate that both CC and CXC types were present in the common ancestor of all vertebrates and that few ancestral genes gave rise to the entire diversity of ligands that we know in current animals. Furthermore, our results indicate that many chemokines, such as CXCL1-7, CXCL16, and CCL25, CCL11/13, and CCL2/7, are uniquely present in mammals, suggesting that the mammal ligand repertoire is substantially more complex than the one observed in other vertebrates.

Regarding noncanonical chemokine-like families, our findings indicate that the TAFA family originated in the ancestor of vertebrates and urochordates; CYTL is a novelty of jawed vertebrates; and CXCL17 is mammal-specific and likely unrelated to canonical chemokines (similar to its controversial putative receptor, GPR35 (39, 56, 57), that is not a canonical chemokine receptor). The CKLF superfamily has a more complex pattern with the presence of few groups in invertebrates and then great expansions occurring at the base of vertebrates. The CKLFSF includes a monophyletic clade (CKLF group I) comprising the original CKLF that binds CCR4, and CMTM1, 2, 3, 5, derived from duplications at the jawed vertebrates stem group. Interestingly, our analysis also revealed that additional molecules not previously considered part of the CKLF super family are closely related to classic members and should be included in it. For example, proteolipid protein 2 (PLP2) belongs to the CKLF I group and is, therefore more closely related to the CKLF with chemokine function than several other CKLFSF members. Similarly, CMTM8 is more closely related to plasmolipin (PLLP) and myelin and lymphocyte protein (MAL) than to any of the classic CKLFSF members. Although this relationship had been proposed based only on sequence similarity (13), our phylogenetic analysis provides additional evidence for it. Therefore, the potential chemokine function of all these additional members should be explored in vitro and in vivo in both vertebrates and invertebrates.

### Most receptors derive from a single gene duplication

Our results clarify the distribution of canonical chemokine receptors in vertebrates (Fig 4), and their evolutionary relationships and identify the pattern of duplication that leads to their origin (Figs 4A and S26). Unlike previous works (80), we identify that atypical receptors do not form a monophyletic group. Specifically, atypical 2 and 4 are part of the canonical clade specifically related to CC-type receptor subclades. Furthermore, we find that the atypical 3 receptors are related to GPR182, supporting previous functional data suggesting that the latter are ACKRs binding CXCL10, 12, and 13 (67). We attribute these differences to our use of wider GPCR sampling and improved methods for phylogenetic inference.

Remarkably, our results do not identify ACKR1 as related to the main chemokine receptors but rather as a divergent clade

vertebrates and a second time in mammals. Homeostatic chemokines (e.g., CXCL12) are generally more ancient than inflammatory ones. CXCL17 and CYTL are mammal- and jawed vertebrate-specific, respectively. TAFA originated in the common ancestor of vertebrates and urochordates, whereas the chemokine-like factor super family is present in invertebrates although key duplications occurred at the base of vertebrates. **(B)** Number of complements for each ligand group at key species nodes is mapped onto the species tree. The number of complements in each group reflects the pattern of duplications. The major increase occurred at the level of jawed vertebrates with canonical chemokines undergoing a second significant increase within placentals. Silhouette images are by Andreas Hejnol (*Xenopus laevis*); Andy Wilson (*Anas platyrhynchos, Taeniopygia guttata*); Carlos Cano-Barbacil (*Salmo trutta*); Christoph Schomburg (*Anolis carolinensis, Ciona intestinalis, Eptatretus burgeri, Petromyzon marinus*); Christopher Kenaley (*Mola mola*); Chuanixn Yu (*Latimeria chalumnae*); Daniel Jaron (*Mus musculus*); Daniel Stadtmauer (*Monodelphis domestica*); Fernando Carezzano (Asteroidea); Ingo Braasch (*Callorhinchus milii*); Jake Warner (*Danio rerio*); Kamil S. Jaron (*Poecilia formosa*); Mali'o Kodis, photograph by Hans Hillewaert (*Branchiostoma lanceolatum*, https://www.phylopic.org/images/719d7b41-cedc-4c97-9ffe-dd8809f85553/branchiostoma-lanceolatum); Margot Michaud (*Canis lupus, Physeter macrocephalus*); NASA (*Homo sapiens sapiens*); Nathan Hermann (*Scophthalmus aquosus*); Ryan Cupo (*Rattus norvegicus*); seung9park (*Takifugu rubripes rubripes*); Soledad Miranda-Rottmann (*Pelodiscus sinensis*, https://www.phylopic.org/images/929fd134-bbd7-4744-987f-1975107029f5/pelodiscus-sinensis); Steven Traver (*Gallus gallus domesticus, Ornithorhynchus anatinus*); Stuart Humphries (*Thunnus thynnus*); T. Michael Keesey (after Colin M. L. Burnett) (*Gorilla gorilla gorilla*); Thomas Hegna (based on picture by Nicolas Gompel) (*Drosophila (Drosophila) mojavensis*); and Yan Wong (*Balanoglossus*).

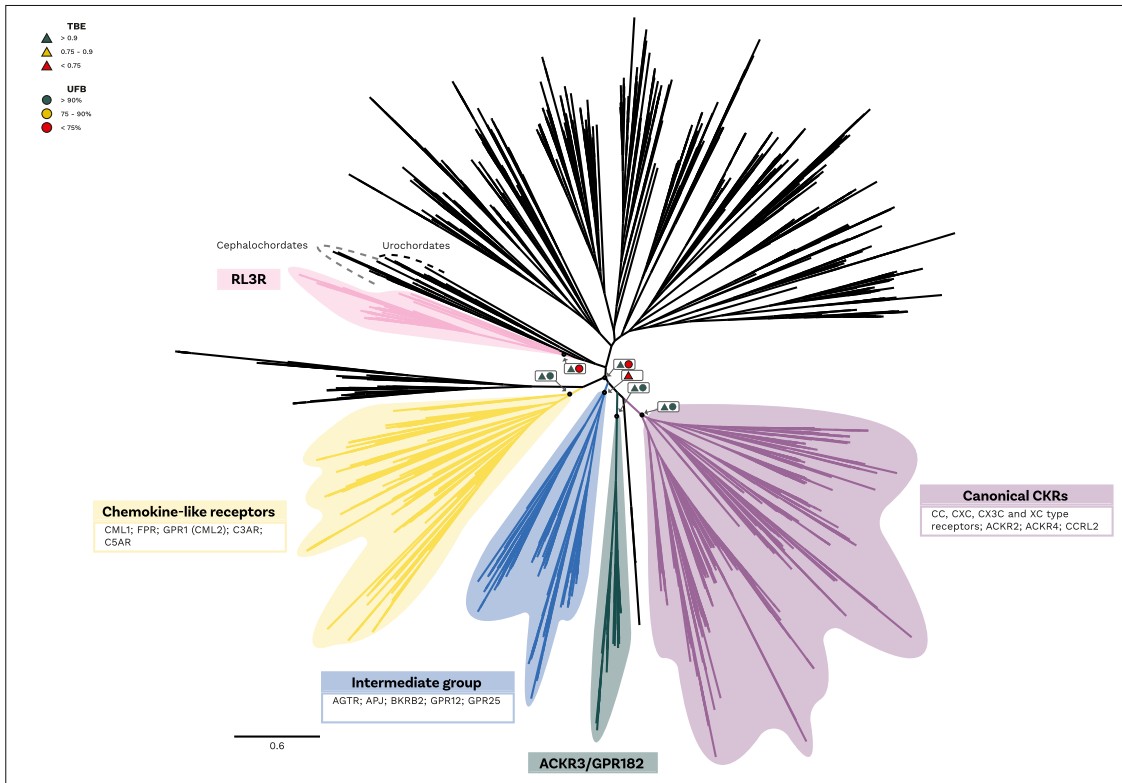

**Figure 3. Phylogeny of receptor groups.**
An unrooted maximum likelihood phylogeny of chemokine receptors. The tree shown is the Transfer Bootstrap Expectation tree including just the chordate specific clade from the Ultrafast Bootstrap tree. Node supports from both Transfer Bootstrap Expectation (triangle) and UFB (circle) shown for equivalent key nodes in boxes with arrows to indicate node. A traffic light colour code is used to indicate the level of support: high (green); intermediate (yellow); and low (red). Key clades highlighted: yellow = chemokine like plus group (CMLplus); blue = intermediate group; green = atypical 3 and GPR182 (ACKR3/GPR182); and pink = relaxin receptors (RL3R). Branches scaled by amino acid substitutions per site.

(Fig S23). To the best of our knowledge, this is the first time this observation has been made. Our current results do not allow us to clarify the evolutionary origin of ACKR1. However, the presence of 7TMD domains suggests that they are GPCRs that independently acquired the ability to bind chemokines. Alternatively, similarly to other genes evolved in the immune system, ACKR1 may have been subjected to strong selective pressures that substantially changed their sequence, obscuring their phylogenetic relationships. The case of ACKR1 being the most distantly related receptor is intriguing as it is one of the most promiscuous chemokine receptors (2, 81) and it has been shown to bind both CC and CXC chemokines (82, 83).

Viral chemokine receptors represent a cryptic group that can bind multiple chemokines (41, 42). Despite their functional similarity to canonical chemokine receptors, viral chemokine receptors' evolutionary origin and distribution remain poorly understood. Our results indicate that viral GPCRs do not form a monophyletic group, suggesting that the ability to encode chemokine-like receptors has evolved independently in multiple viruses, including cytomegaloviruses and poxviruses. The placement of viral sequences within an otherwise vertebrate-specific clade supports the hypothesis that viruses acquired these genes through non-vertical inheritance. Given the paraphyly of viral receptors, this appears to have occurred multiple times. However, there are significant uncertainties details of viral chemokine receptors' evolution.

Our analysis reveals that the clade comprising apelin receptors, angiotensin receptors, bradykinin receptors, and orphan GPCRs (shown in Figs 3 and 4 and S24, S25, and S26) is closely related to chemokine receptors. This finding partially supports previous studies (68) that suggested a gene duplication event gave rise to both chemokine receptors and angiotensin receptors. Interestingly, we found that single gene duplication in the vertebrate stem group led to the emergence of canonical receptors and atypical 2,3,4, GPR182, chemokine-like receptors, FPR, the intermediate group, and many other known and orphan GPCRs including the controversial putative CXCL17 receptor GPR35. These findings suggest that two rounds of genome duplication (84, 85) played a role in the expansion of GPCR gene families. Future research will focus on investigating the functions of the orphan genes and many-to-one orthologs discovered in urochordates. This will provide further insight into the evolution and diversification of GPCR families in vertebrates.

## The molecular assembly of the chemokine system

In this work, we explored the evolution of both ligand and receptor components of the chemokine signaling system, including noncanonical molecules with either chemokine-like function or

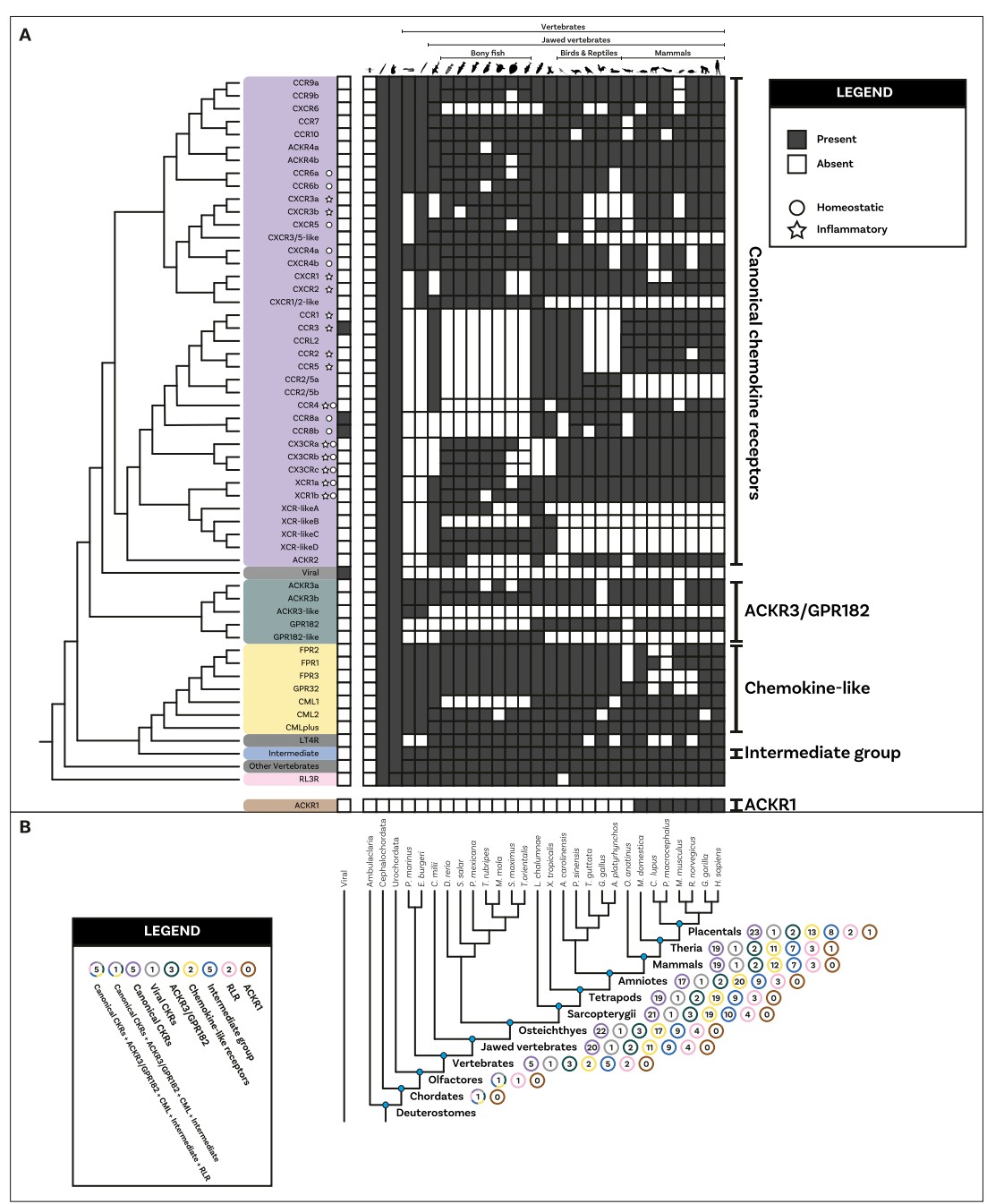

**Figure 4. Distribution and duplication patterns of receptor groups.**
**(A)** Presence of all receptor groups are mapped onto a species tree. Gene trees and duplication events are based on the gene tree to species tree reconciliation analyses. The nomenclature for genes is primarily based on human chemokines. The canonical chemokines had five paralogs present in the vertebrate common ancestor. These undergo a heterogeneous pattern of duplication throughout vertebrates with different paralogs duplicating different number of times and in different groups of species. Chemokines that have been classically described as having either homeostatic or inflammatory function are indicated with a circle or a star respectively. The classification used here was based on reference 7. **(B)** Number of complements for each receptor group at key species nodes is mapped onto the species tree. The number of complements in each group reflects the pattern of duplications. The chemokine groups diverged in the vertebrate stem group. The major expansion occurred at the level of jawed vertebrates with canonical chemokine receptors, the chemokine-like receptor plus group and intermediate groups increasing in copy number. Canonical chemokine underwent another small subsequent increase within placentals. Silhouette images are by Andreas Hejnol (*Xenopus laevis*); Andy Wilson (*Anas platyrhynchos*, *Taeniopygia guttata*); Carlos Cano-Barbacil (*Salmo trutta*); Christoph Schomburg (*Anolis carolinensis, Ciona intestinalis, Eptatretus burgeri, Petromyzon marinus*); Christopher Kenaley (*Mola mola*); Chuanixn Yu (*Latimeria chalumnae*); Daniel Jaron (*Mus musculus*); Daniel Stadtmauer (*Monodelphis domestica*); Fernando Carezzano (Asteroidea); Ingo Braasch (*Callorhinchus milii*); Jake Warner (*Danio rerio*); Kamil S. Jaron (*Poecilia formosa*); Mali'o Kodis, photograph by Hans Hillewaert (*Branchiostoma lanceolatum*, https://www.phylopic.org/images/719d7b41-cedc-4c97-9ffe-dd8809f85553/branchiostoma-lanceolatum); Margot Michaud (*Canis lupus, Physeter macrocephalus*); NASA (*Homo sapiens sapiens*); Nathan Hermann (*Scophthalmus aquosus*); Ryan Cupo (*Rattus norvegicus*); seung9park (*Takifugu rubripes rubripes*); Soledad Miranda-Rottmann (*Pelodiscus sinensis*, https://www.phylopic.org/images/929fd134-bbd7-4744-987f-1975107029f5/pelodiscus-sinensis); Steven Traver (*Gallus*

sequence similarity. Our analysis suggests that the canonical chemokine signaling evolved in the vertebrate stem group (about 500 Mya) likely because of the two rounds of genome duplication that gave rise to many vertebrate novelties ([84], [85]). We found that the ancestral vertebrate repertoire included orthologs of both major ligand groups (CXCL and CCL) and both CCR and CXCR receptors and noncanonical components such as TAFA and CKLFSF ligands, and the receptors Atypical 3 and GPR182 ([Fig 5]). The distribution of ligands and receptors in the ancestor of all vertebrates seems to confirm the hypothesis that the ancestral function of chemokines was homeostatic (e.g., CXCL12, CXCL14) with inflammatory functions arising from recent duplications (e.g., CXCL5, CXCL6), potentially reflecting a rapid evolution induced by the selective pressure of new pathogens ([7]). Chemokine ligand and receptor genes are known to cluster on specific chromosomes ([7]), consistent with the hypothesis that they may be the result of the combination of en bloc duplication followed by tandem duplications ([47], [63], [64]). Because of limited high-quality genomes, syntenic patterns of chemokine genes described so far are based primarily on humans and a handful of other species ([47], [63], [64]), hampering our understanding of the level of conservation of these syntenic patterns. Conversely, our large-scale phylogenetic analyses encompassed many species. We uncovered several phylogenetic relationships that are consistent with known syntenic patterns in human, providing stronger evidence for their evolutionary relationship.

The evolutionary history of canonical components includes several examples of known ligand–receptor pairs following a corresponding pattern of origin and temporal dynamics of duplications. This is true, for example, for the ancient homeostatic CXCL12 ligand and its corresponding receptors CXCR4 and ACKR3, that all originated in early vertebrates ([7]). The early origin and conservation of CXCR4 and CXCL12 in the ancestor of vertebrates is interesting as this pair plays a key role in the migration of neural crest cells ([86])—a key vertebrate innovation ([87]). This, combined with the fact that homeostatic chemokine ligands/receptors tend to be restricted to monogamous pairing ([2], [65]) suggests that homeostatic chemokine pairings are more ancient and conserved being in single copy throughout much of the vertebrates. Contrastingly, inflammatory chemokine pairings are more promiscuous, and this could be linked to the more recent duplications in the genes, such as for CCL2/7/8/11/13 ([Fig 2A]) and their receptors CCR1/2/3/4/5 ([Fig 4A]). For many of the noncanonical components, however, the ligand–receptor interactions are largely unclear, and their pattern throughout vertebrate evolution remains to be explored. Overall, our results indicate that three waves of molecular innovation in the vertebrates, jawed vertebrates, bony fishes, and mammal stem group increased the chemokine system's molecular complexity ([Fig 5]), allowing for the fine-tuning present in modern-day animals.

# Materials and Methods

## Data mining and dataset assembly

We collected 64 proteomes from 25 vertebrates, six chordates, and 33 other animals covering the whole animal tree (Table S1). BUSCO v4.0.6 ([88], [89]) and the metazoa_odb10 set of 954 genes were used to evaluate their completeness (Table S1).

To identify potential homologs of canonical chemokines, TAFA chemokines, and CYTL1, we used 207 curated sequences that we obtained from SwissProt ([90], [91]) as seeds for an initial BLASTP ([51], [53]) with $E$-value < $10^{-10}$. To identify putative chemokines in cyclostomes, the lamprey *Petromyzon marinus* ([92]), and the hagfish *Eptatretus burger* ([93] *Preprint*), we loosened the $E$-value to 0.05. Where putative chemokine sequences were found for one cyclostome species but not the other, those sequences were used to search again the other species. Furthermore, to investigate the presence of ligands outside vertebrates, we performed an additional BLASTP on invertebrate proteomes with an even looser $E$-value (0.1) and collected only up to five hits. This provided 18 initial candidate homologs spanning multiple invertebrate phyla. Further characterisation of these invertebrate sequences, through BLASTP versus SwissProt, protein domains search with InterProScan ([94], [95]), position in CLANS analysis (see below) and, where necessary, multiple sequence alignments, led us to retain only one urochordate sequence as a putative TAFA homolog (see the Supplementary results section for details).

To identify homologs for the CKLF superfamily, we used 21 SwissProt-reviewed sequences. In addition to the BLASTP search, we used a position-specific iterative BLAST (PSI-BLAST) ([52]) with an $E$-value threshold of <$10^{-10}$. Using this approach, we identified a total of 590 putative homologs, including 186 from invertebrates.

We used BLASTP using 178 manually annotated receptor sequences from SwissProt as query sequences for the chemokine receptors. This includes all human canonical and ACKRs ([96]). We also collected eight viral sequences with chemokine receptor activity from UniProt ([97]) and performed a second BLASTP. We extracted all BLAST hits with $E$-values < $10^{-10}$ and used Phobius ([98]) to predict their transmembrane domain structure. Only sequences with five to eight transmembrane domains were kept. Hit sequences were annotated by their top five BLAST hits against SwissProt. All hits from both BLASTs were merged and filtered by cd-hit ([99], [100]) to remove redundant sequences at the 95% similarity threshold. This resulted in 7,157 putative chemokine GPCR sequences.

## Identification of subgroups with CLANS

We used CLANS ([54], [55]) with default parameters and different $P$-values (i.e., stringency values) to visualize the relationships between subgroups of ligands and receptors. We assessed the similarity and interrelationships between different clusters by gradually

*gallus domesticus, Ornithorhynchus anatinus*); Stuart Humphries (*Thunnus thynnus*); T. Michael Keesey (after Colin M. L. Burnett) (*Gorilla gorilla gorilla*); Thomas Hegna (based on picture by Nicolas Gompel) (*Drosophila (Drosophila) mojavensis*); and Yan Wong (*Balanoglossus*).

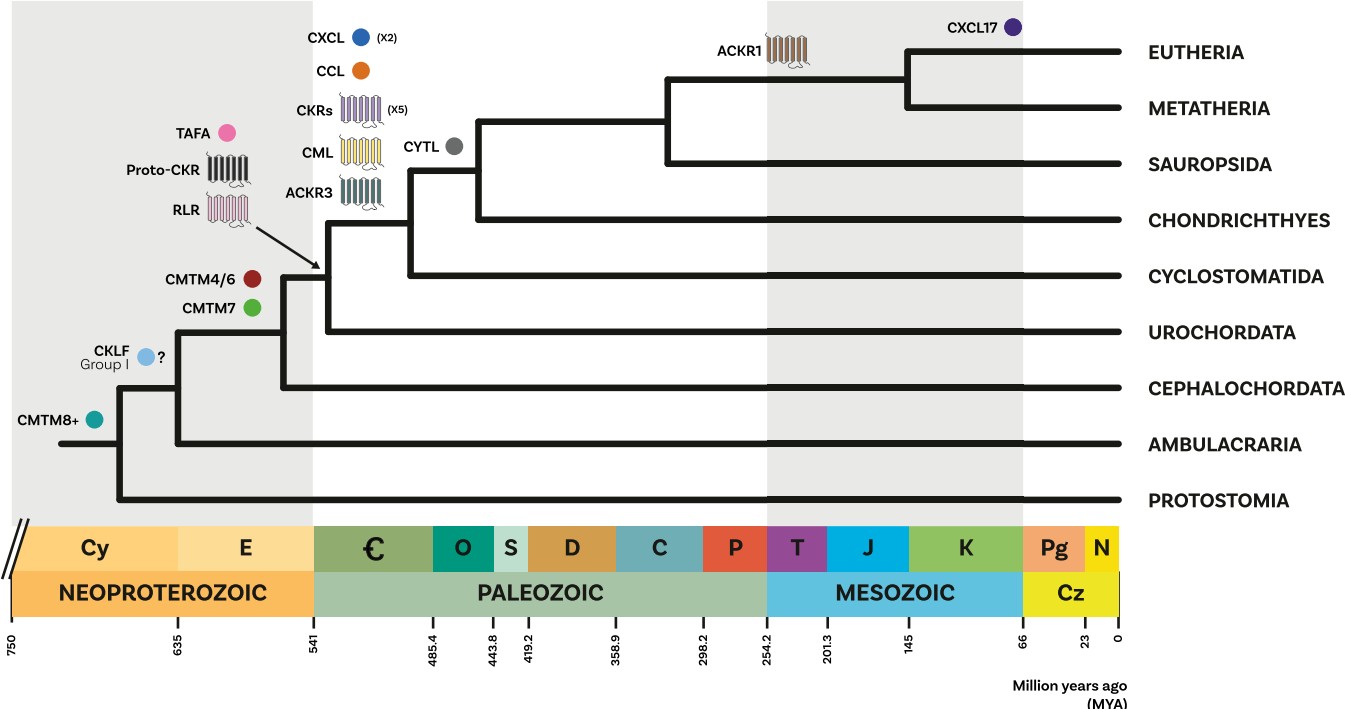

**Figure 5. Summary of the evolution of ligands and receptors.**
A summary diagram of the evolution of the different chemokine system components. A simplified phylogenetic tree of species is shown, calibrated to time according to reference 73 for Deuterostomia and Bilateria nodes and reference 74 for all other nodes. Circles represent ligand groups, and seven transmembrane domain structure icons represent GPCR groups. Icons are colour-coded by group, and placed adjacent to the branch in the species tree where they first appear. X2 and X5 indicate the number of paralogs present for CXCL ligand group and the canonical CKR groups, respectively, on the branch where they first appear. Question mark refers to the uncertainty regarding the origin of the chemokine-like factor group I in jawed vertebrates or deuterostome stem group (see Fig 2). Geological column is shown along the bottom, in accordance with the ICS International Chronostratigraphic Chart (75).

relaxing the *P*-value threshold (Figs S1, S2, and S13). In addition, we annotated each cluster using gene annotations for key species *Homo, Mus, Gorilla, Gallus, Anolis,* and *Danio*. In the case of the receptors, to improve the cluster annotation all human Class-A GPCRs (excluding olfactory receptors) from GPCRdb (101) were added to the dataset and the eight seed viral chemokine receptors from UniProt (97).

### Alignment and phylogenetic analysis

#### Alignment
All ligand and receptor sequences were aligned using MAFFT (102, 103) with the –auto setting and using trimAl (104) to remove positions with >70% gaps.

#### Gene trees
All gene alignments were analysed using IQTREE2 (105); the model test algorithm (106) was used to select the best substitution model for each analysis. The best models selected by IQTREE2 for each set are listed in Table S2 (for receptors we manually selected GTR20+F+G4 as the model as it was a large dataset). Nodal support was estimated using 1,000 UFB (58, 59) replicates. All analyses were repeated to run 100 nonparametric bootstrap repeats to calculate

nodal support with TBE which is specifically designed to account for phylogenetic instability (60).

For the receptors, because of the high computational burden of running TBE analyses on sequence-dense datasets, we first analysed the full set of 3,026 sequences connected in CLANS at a *P*-value of $< 1 \times e^{-50}$ using UFB (Fig S25). Then, we extracted the chordate-specific clade sequences, including all chemokine receptor groups and their immediate outgroups, to analyse using TBE.

#### Gene tree–species tree reconciliation
To understand the pattern of duplication and the evolution of gene complement we used GeneRax (61). GeneRax requires a gene tree that was obtained as described above and a species tree that we constructed manually using publicly available information. In the instances where the genes tree contained polytomies, we used ETE3 (107) to solve them. The undated DL mode and the closest approximation of the best-fitting substitution model were used for each alignment. To track the evolution of sub-lineages within each group, we used annotated sequences of key species (e.g., *Homo sapiens* and *Mus musculus*) as reference. For the receptors, we used only the chordate-specific clade subtree and sequences because of the computational burden of running GeneRax on a high number of sequences. For species tree–gene tree reconciliation, we treat the viral sequences as human sequences.

## Supplementary results

### Exploration of candidate invertebrate homologs to canonical chemokines and related molecules

To explore the possibility of finding canonical chemokines and/or related molecules (TAFA and CYTL) outside of vertebrates, we used BLASTP with a loose *E*-value threshold of 0.1 to search 39 invertebrate proteomes (see Table S2). 18 candidate sequences were collected and explored further.

From the clustering analysis in CLANS (Figs 1A and S1A–D and Supplementary File 1 in the GitHub repository: Roberto-Feuda-Lab/Chemokine2023 (github.com)), it became apparent that four sequences were candidate TAFAs, whereas the remaining 14 sequences connected loosely to the canonical chemokines. We then performed BLASTP (51, 53) versus the curated SwissProt dataset (90, 91) and collected the first five hits. Furthermore, we used InterProScan (94, 95) to identify protein signatures. See Supplementary File 3 in the GitHub repository: Roberto-Feuda-Lab/Chemokine2023 (github.com) for a summary of all these results for all sequences.

Regarding the canonical chemokines, only three sequences received annotations related to chemokines from the BLASTP versus SwissProt. These were the following: one brachiopod sequence (*Lingula unguis*) as candidate CCL24, one cnidarian (*Clytia hemisphaerica*) sequence as candidate CCL3, and one echinoderm (*Acanthaster planci*) sequence as candidate CXCL10. Although none of these sequences were categorised as chemokines with InterProScan, we anyway decided to look at them further. First, we noted how all three sequences were significantly longer than their counterparts in vertebrates. Second, none of the three sequences possessed a signal peptide, as calculated with SignalP 6.0 online tool (https://services.healthtech.dtu.dk/service.php?SignalP-6.0), which is expected from secreted proteins such as chemokines (108). Finally, we anyway tried to align the sequences (MAFFT –auto) with their respective candidate relatives and found poor conservation (Figs S13, S14, and S15). The lack of evidence for being true chemokines, led us to discard all invertebrate candidates for further analyses on canonical chemokines.

The four candidate invertebrate TAFA sequences all belong to the urochordate *Ciona intestinalis*. One sequence was annotated as TAFA by both SwissProt and InterProScan, whereas the other three appear to be prolyl hydroxylases (see Supplementary File 3 in the GitHub repository: Roberto-Feuda-Lab/Chemokine2023 (github.com)). We anyway studied all four sequences further and found that the only one to possess a signal peptide is the sequence that received TAFA annotation. Moreover, the other three sequences appear to be too long and poorly aligned with vertebrate TAFAs (Fig S17). The TAFA annotated sequence was of correct length and showed sequence conservation in the alignments (Figs S17 and S18). Interestingly, it possesses 8 of the 10 typical cysteine residues of TAFA1–4 and the two missing cysteines are the same ones missing in TAFA5. Considering that TAFA5 is the sister group to TAFA1–4 and that the urochordate sequence places itself as orthologous to all TAFAs (see main text Results), it is reasonable to conclude that the ancestral TAFAs possessed eight cysteine residues and that the additional cysteines are a novelty of the TAFA1–4 lineage.

Taken together, these results show that whereas canonical chemokines are indeed a vertebrate innovation, TAFA "chemokines" likely originated in the ancestor of vertebrates and urochordates.

### Exclusion of some sequences from the CKLFSF dataset

The data mining through BLASTP and PSI-BLAST (52) provided numerous candidate CKLFSF homologs both in vertebrates and in invertebrates and these were all included in a clustering analysis with CLANS (Figs 1C and S2A–D). Two main clusters emerged, and we called them "CKLF I" and "CKLF II." Whereas CKLF I was vertebrate specific, the CKLF II included multiple invertebrate sequences. Although the two main clusters are well defined already at *P*-values of $\sim 1 \times 10^{-20}$ (Fig S2B and Supplementary File 2 in the GitHub repository: Roberto-Feuda-Lab/Chemokine2023 (github.com)), they only connect to each other at $1 \times 10^{-15}$. At this *P*-value, four additional sequences connected loosely to the CKLF II cluster. These were three echinoderm sequences (all from *Stichopus japonicus*) and one sequence from the placozoan *Trichoplax adhaerens*. The latter is the only non-bilaterian sequence collected from the original BLASTs. These sequences not only joined the CKLFSF cluster at the limit threshold, but also connected only to sequences that were already marginal, therefore being only indirectly connected to the core of the cluster. Like above, we examined the sequences with a BLAST versus SwissProt and with InterProScan (see Supplementary File 3 in the GitHub repository: Roberto-Feuda-Lab/Chemokine2023 (github.com)). The evidence in favour of keeping these sequences was scant (see details in Supplementary File 3 in the GitHub repository: Roberto-Feuda-Lab/Chemokine2023 (github.com)), and we decided to exclude them from downstream phylogenetic analyses. The CKLFSF dataset therefore did not include any non-bilaterian sequences, although multiple bilaterian invertebrate phyla were represented.

## Data Availability

Supplementary material and raw output files for all the analyses described in this article are available at the GitHub repository: Roberto-Feuda-Lab/Chemokine2023 (github.com). All data are also deposited in the Zenodo repository: (109).

## Supplementary Information

## Acknowledgements

This work is supported by a University Research Fellowship (UF160226) to R Feuda. A Aleotti is supported by a Research Grant from the Royal Society to R Feuda (RGF\R1\181012). M Goulty is supported by a PhD Scholarship from the University of Leicester. C Lewis is supported by a BBRSC MIBPT fellowship. This research used the ALICE High-Performance Computing Facility at the University of Leicester.

## Author Contributions

A Aleotti: conceptualization, data curation, formal analysis, investigation, and writing—original draft, review, and editing.
M Goulty: conceptualization, data curation, formal analysis, investigation, methodology, and writing—original draft, review, and editing.
C Lewis: data curation, formal analysis, and visualization.
F Giorgini: conceptualization, supervision, methodology, and writing—original draft, review, and editing.
R Feuda: conceptualization, formal analysis, supervision, funding acquisition, project administration, and writing—original draft, review, and editing.

## Conflict of Interest Statement

The authors declare that they have no conflict of interest.

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
