## [Reviewer comments · Life Science Alliance]

Life Science Alliance

The origin, evolution and molecular diversity of the chemokine system

Alessandra Aleotti, Matthew Goulty, Clifton Lewis, Flaviano Giorgini and Roberto Feuda

DOI: <https://doi.org/10.26508/lsa.202302471>

Corresponding author(s): Roberto Feuda (University of Leicester)

Review Timeline:

Submission Date:	2023-11-06
Editorial Decision:	2023-12-01
Revision Received:	2023-12-15
Accepted:	2023-12-18

Transaction Report:

Please note that the manuscript was reviewed at *Review Commons* and these reports were taken into account in the decision-making process at *Life Science Alliance*.

Review
COMMONS

Review #1

The authors describe a broad-scale phylogenetic survey of chemokine-related ligand and receptors from representative vertebrates, invertebrates, and viruses. They collect ligand and receptor sequences from available genome sequences, and use phylogenetic and CLANS analysis to group these into similar gene types. They then overlay these onto a validated species phylogeny in order to evaluate relationships of orthology and paralogy to pinpoint gene duplication and loss events. They carry out these analyses for canonical chemokine ligands receptors and for other closely related protein families. They conclude that the canonical chemokine system is restricted to vertebrates but that closely related ligands and receptors can be found in invertebrate chordates. More divergent but related gene systems are found in more distant invertebrates. They define more limited expansions of some ligand-receptor systems in certain jawed vertebrate groups and specifically in mammals.

Overall, the paper addresses a complex and important system of signaling proteins with a rigorous and comprehensive set of analyses. The finding will be of interest to a diverse group of scientists. My comments listed below mainly consist of suggestions to help clarify the presentation.

1. Pg 2, Lns 21-24: The canonical and non-canonical chemokine subclasses are introduced in the abstract without definition. A very brief explanation would be useful.
2. Some general contexts of chemokine functions are listed, including inflammation and homeostasis. A little more detail of how these signals are used and the molecular consequences of signaling may be useful in the introduction to set the biological context of the analysis (e.g., how do the signals regulate homeostasis?).
3. It may help to summarize the known chemokine and chemokine-related gene systems in some type of table at the beginning of the results. This could serve as a convenient reference to guide the reader through the more detailed results. The manuscript addresses a complex set of ligands and receptors with names that may be confusing to the non-expert.
4. Pg 5, Ln 98: Fig 1C is introduced before Fig 1B. Can the panels be switched or the descriptions be rearranged?
5. Cytokine and chemokine ligands are small proteins that diverge quickly in different species and are difficult to identify in divergent genomes even within vertebrates. Conclusions about the absence of these types of factors are notorious for being disproven in subsequent analyses. Some discussion of what may have been missed in the survey for homologs (or reasons to think that ligands were not missed) would be useful in the Discussion.

This paper presents a thorough analysis of chemokines and related gene systems across a wide phylogenetic landscape. The authors have expertise in these gene families and in the techniques that they use to identify and relate family members. The chemokines are an important set of signals that are used across several biological systems. These findings will be of wide interest to immunologists, neurobiologists, developmental and evolutionary biologists.

Review #2

This paper applies phylogenetic clustering methods to a large taxonomical sampling to interrogate the relationship between canonical and non-canonical chemokine ligands and receptors. The results suggest that 1) unrelated proteins evolved "chemokine-like" ligand function multiple times independently; and 2) all the canonical and non-canonical chemokine receptors (except ACKR1) originated from a single duplication in the vertebrate stem group, which also gave rise to many GPCRs. In addition, the authors characterized the complement of canonical and non-canonical components in the common ancestor of vertebrates and identified several other ligands and receptors with potential chemokine related properties.

****Comments:****

1. There are many places in the paper, too many to list, where the authors refer to chemokine receptors but call them 'chemokines'.
2. In Figure 1, CX3CL is referred to as 'X3CL'
3. CXCL17 was originally reported to be chemokine-like based on sequence threading methods. The authors

refer to a 2015 paper indicating that it has chemokine-like activity at GPR35, which had been renamed provisionally CXCR8. To my knowledge that result was not based on direct binding data but inferred from a functional response. Moreover, to my knowledge it has not been independently confirmed. Instead there is a recent paper in JI from the Pease lab showing extensive experimental results that fail to demonstrate CXCL17 activity at GPR35. This uncertainty regarding a potential mistake in the literature should be addressed and integrated in the points made about CXCL17 being an outlier.

4. Can the authors use alpha fold to address whether any of these non-canonical molecules actually is predicted to fold like a chemokine? More generally, based on the paper's analysis, how do the authors propose to define a chemokine? It is well-accepted that chemokines are defined by structure, not function (e.g. limited truncation of any chemokine abrogates activity, but it is still a chemokine structurally, not semantically, folds like a chemokine, aligns with other chemokines).

5. Chemokine genes are found on many human chromosomes with large clusters on chromosome 2 and 17. Can the authors address the syntenic relationships phylogenetically?

6. The authors indicate that 'CXCL8 is present in all jawed vertebrates except in the cartilaginous fishes lineage'. However, they should point out that CXCL8 is not represented in mice. The notion that the repertoire of chemokine and chemokine receptor genes can be different in even closely related species as well as in individuals of the same species is well-documented but not mentioned here.

7. The analysis suggests that chemokine gene repertoires start small and grow non-linearly to 45 in mammals. However DeVries et al (JI 2005) published that zebrafish have the most chemokines, 63, and chemokine receptors, 24. Do the authors disagree? This should be addressed.

8. Did the authors find any species in which a chemokine/chemokine receptor pair are not found together? That is, if the system is irreducibly complex, requiring both a ligand and receptor, the probability of both genes arising simultaneously is essentially zero. So how do the authors theorize that such a system actually arose, and is there any evidence in their data set for convergence of separately evolved ligand and receptor?

9. Line 180, 181 and elsewhere: GPCR1 and GPCR33 should be GPR1 and GPR33

10. Line 185: ACKR1 exceptionalism is noted, but there is no discussion of the remarkable structure-function paradox that the most distantly related chemokine receptor is also the most highly promiscuous receptor, binding many but not all CC and CXC chemokines with high affinity.

11. Line 196: the viral receptors cluster with the vertebrate receptors, suggesting that the viruses captured the receptor gene from the host. Authors might mention this obvious point regarding origins, and discuss how it relates to the monophyly and paraphyly that emerges from the phylogenetic analysis.

12. Any discussion of chemokine-like convergent evolution presupposes that the activity is real and actually occurs in vivo. The authors should make clear to what extent the existing literature supports this. As mentioned above, CXCL17 interaction with GPR35 has been challenged in vitro and has never been demonstrated to occur in vivo. To what extent is the same limitation a problem in considering co-evolution of the other non-canonical chemokines? I agree that classification based solely on function is inappropriate, but so is phylogenetic analysis without direct knowledge of in vivo function. It is not feasible to address this in a phylogenetic analysis, but there ought to be at least one species in which the non-canonicals have been rigorously shown to act at specific receptors in vivo before grouping them with the canonicals in a co-evolutionary sense.

13. The discussion of homeostatic vs inflammatory chemokine/receptors in the last section of the Discussion would be enhanced by pointing out that the chemokine specificities are numerically totally different for these two groupings, homeostatics tending to have monogamous ligand-receptor relationships and inflammatory being highly promiscuous.

Much of the paper's results are confirmatory of previous work based on less extensive sequence analysis. One could say more generally that unrelated chemical forms, not just unrelated proteins, have chemokine-like ligand function. For example leukotriene B4 is a powerful leukocyte chemoattractant for neutrophils working through a GPCR. That proteins might also independently evolve common functions does not add insight beyond what is already appreciated. The notion that chemokine receptors have a common ancestor is also generally accepted and that ACKR1 is an outlier is already appreciated. The present work adds phylogenetic and statistical precision to these points.

Our study investigates the origin and evolution of the chemokine system, which plays a pivotal role in numerous biological functions, including immune response and neuronal development. Despite the extensive research devoted to its functional aspects (with over 300,000 works on PubMed), the origin and evolution of this system have remained enigmatic due to the number of canonical and non-canonical components involved and the challenges associated with reconstructing phylogenies for short sequences. To address these difficulties, we investigate the entity of ligands and receptors (both canonical and non-canonical) using a large set of genomes (64 species from 19 phyla and a newly sequenced Hagfish genome) and state-of-the-art phylogenomic tools. Our key finding includes:

- 1) Evidence that multiple independent molecules acquired the ability to interact with chemokine receptors.
- 2) The origin of all receptors, except for the atypical 1 receptor, through a single gene duplication event in the vertebrate stem group.
- 3) The identification of three distinct waves of molecular diversification within the system in vertebrates.
- 4) The paraphyly of viral receptors.
- 5) The identification of additional ligands and receptors with potential chemokine functions will open exciting avenues for future exploration.

Collectively, these results substantially clarify the molecular evolution and diversity of the chemokine system. They also challenge many previously held conclusions (as elaborated in our response to Reviewer #2 Significance statement). Given the central role of chemokines in many biological processes, we anticipate that our discoveries will significantly impact diverse scientific domains, extending from evolutionary biology to immunology (see Reviewer #1 Significance statement), resulting in a high number of citations.

Reviewer #1 (Evidence, reproducibility and clarity (Required)):

The authors describe a broad-scale phylogenetic survey of chemokine-related ligand and receptors from representative vertebrates, invertebrates, and viruses. They collect ligand and receptor sequences from available genome sequences, and use phylogenetic and CLANS analysis to group these into similar gene types. They then overlay these onto a validated species phylogeny in order to evaluate relationships of orthology and paralogy to pinpoint gene duplication and loss events. They carry out these analyses for canonical chemokine ligands receptors and for other closely related protein families. They conclude that the canonical chemokine system is restricted to vertebrates but that closely related ligands and receptors can be found in invertebrate chordates. More divergent but related gene systems are found in more distant invertebrates. They define more limited expansions of some ligand-receptor systems in certain jawed vertebrate groups and specifically in mammals.

Overall, the paper addresses a complex and important system of signaling proteins with a rigorous and comprehensive set of analyses. The finding will be of interest to a diverse group of scientists. My comments listed below mainly consist of suggestions to help clarify the presentation.

1. Pg 2, Lns 21-24: The canonical and non-canonical chemokine subclasses are introduced in the abstract without definition. A very brief explanation would be useful.

We've included a brief description of "non-canonical" components in the abstract (lines 21-24). These non-canonical components fall into at least one of three categories: 1) molecules with sequence similarities to canonical components, 2) those that bind to a canonical component (either ligand or receptor), 3) those involved in chemokine-like functions, such as chemoattraction. More comprehensive explanations and examples of these non-canonical components are provided in the Introduction section.

2. Some general contexts of chemokine functions are listed, including inflammation and homeostasis. A little more detail of how these signals are used and the molecular consequences of signaling may be useful in the introduction to set the biological context of the analysis (e.g., how do the signals regulate homeostasis?).

We have added at the beginning of the introduction (lines 39 – 46) some details of how chemokine signalling typically occurs at a mechanistic level. We also provided few examples of homeostatic functions regulated by chemokine signalling and clarified different expression strategies for inflammatory versus homeostatic chemokines.

3. It may help to summarize the known chemokine and chemokine-related gene systems in some type of table at the beginning of the results. This could serve as a convenient reference to guide the reader through the more detailed results. The manuscript addresses a complex set of ligands and receptors with names that may be confusing to the non-expert.

We agree with the reviewer on this and moved Table S1 to the main text (now Table 1). This table contains all the information on ligands, receptors, and relative citations (lines 741-744).

4. Pg 5, Ln 98: Fig 1C is introduced before Fig 1B. Can the panels be switched or the descriptions be rearranged?

We have switched the panels in Figure 1. Now, Figure 1A and 1B refer to CLANS analyses and Figure 1C and 1D refer to phylogenetic trees of ligand groups. We have corrected all the references in the main text and in Figure 1 caption. Now the panels are mentioned in the correct alphabetical order within the text.

5. Cytokine and chemokine ligands are small proteins that diverge quickly in different species and are difficult to identify in divergent genomes even within vertebrates. Conclusions about the absence of these types of factors are notorious for being disproven in subsequent analyses. Some discussion of what may have been missed in the survey for homologs (or reasons to think that ligands were not missed) would be useful in the Discussion.

We concur with the reviewer's observation, and we used three distinct strategies to address the issue:

1. E-value Threshold Adjustment: Initially, we utilized a relatively low e-value threshold of $<e^{-10}$ for sequence identification. In specific instances, particularly among basal vertebrates or invertebrates, we relaxed this threshold to 0.05 or 0.1. This adaptive approach enabled us to pinpoint 18 potential sequences outside the realm of vertebrates. However, these sequences were subsequently excluded from our analysis due to their absence of specific motifs (see Supplementary Results).

2. Incorporation of Multiple Species: Whenever feasible, we included multiple species within each taxonomical group. This approach was adopted to mitigate species-specific variations in terms of presence and absence of the targeted sequences. By diversifying our species selection, we aimed to reduce potential biases.

3. Consideration of Taxonomic and BUSCO Completeness: In the selection of species, we placed emphasis not only on taxonomic representation but also on BUSCO gene completeness, as detailed in Table S1. In cases where certain species exhibited a low BUSCO score, we also included closely related species within the same taxonomic group. This decision increased the likelihood of identifying homologous sequences within that specific taxonomic cluster.

These three strategies collectively contribute to a more robust and comprehensive approach to address the challenges associated with the bioinformatic identification of canonical and non-canonical chemokines. We briefly mentioned the technical difficulty of working with short sequences in our Introduction (lines 75-76).

Reviewer #1 (Significance (Required)):

This paper presents a thorough analysis of chemokines and related gene systems across a wide phylogenetic landscape. The authors have expertise in these gene families and in the techniques that they use to identify and relate family members. The chemokines are an important set of signals that are used across several biological systems. These findings will be of wide interest to immunologists, neurobiologists, developmental and evolutionary biologists.

We thank reviewer 1 for their comments – they have been very valuable to improve our manuscript.

Reviewer #2 (Evidence, reproducibility and clarity (Required)):

This paper applies phylogenetic clustering methods to a large taxonomical sampling to interrogate the relationship between

canonical and non-canonical chemokine ligands and receptors. The results suggest that 1) unrelated proteins evolved "chemokine-like" ligand function multiple times independently; and 2) all the canonical and non-canonical chemokine receptors (except ACKR1) originated from a single duplication in the vertebrate stem group, which also gave rise to many GPCRs. In addition, the authors characterized the complement of canonical and non-canonical components in the common ancestor of vertebrates and identified several other ligands and receptors with potential chemokine related properties.

Comments:

1. There are many places in the paper, too many to list, where the authors refer to chemokine receptors but call them 'chemokines'.

We have corrected this oversight throughout the manuscript.

2. In Figure 1, CX3CL is referred to as 'X3CL'

We have corrected this. Now CX3CL is referred to correctly in Figure 1. We also found that it was incorrectly spelled in Figure 2 as well and corrected it there too.

3. CXCL17 was originally reported to be chemokine-like based on sequence threading methods. The authors refer to a 2015 paper indicating that it has chemokine-like activity at GPR35, which had been renamed provisionally CXCR8. To my knowledge that result was not based on direct binding data but inferred from a functional response. Moreover, to my knowledge it has not been independently confirmed. Instead there is a recent paper in JI from the Pease lab showing extensive experimental results that fail to demonstrate CXCL17 activity at GPR35. This uncertainty regarding a potential mistake in the literature should be addressed and integrated in the points made about CXCL17 being an outlier.

We thank the reviewer for pointing this out. To account for this suggestion, we have modified the text as follows:

Lines 105-108: "The distinction between CXCL17 and all other canonical chemokines is consistent with our receptor results showing that the potential receptor for CXCL17, GPR35 (41), is also not within the canonical chemokine receptor group (see below). Although it is important to note that recent studies fail to demonstrate CXCL17 activity at GPR35 (42, 43)."

Lines 240-241: "Another orphan GPCR, GPR35, had been proposed as a potential chemokine receptor (41); however, this was later questioned (42, 43) and GPR35 is still generally considered orphan (55–57)."

Lines 312-315: "CXCL17 is mammal-specific and likely unrelated to canonical chemokines (similar to its controversial putative receptor, GPR35 (41-43), that is not a canonical chemokine receptor)."

References:

[41] J. L. Maravillas-Montero, et al., Cutting Edge: GPR35/CXCR8 Is the Receptor of the Mucosal Chemokine CXCL17. The Journal of Immunology 194, 29–33 (2015).

[42] S.-J. Park, S.-J. Lee, S.-Y. Nam, D.-S. Im, GPR35 mediates Iodoxamide-induced migration inhibitory response but not CXCL17-induced migration stimulatory response in THP-1 cells; is GPR35 a receptor for CXCL17? British Journal of Pharmacology 175, 154–161 (2018).

[43] N. A. S. B. M. Amir, et al., Evidence for the Existence of a CXCL17 Receptor Distinct from GPR35. The Journal of Immunology 201, 714–724 (2018).

[55] S. Xiao, W. Xie, L. Zhou, Mucosal chemokine CXCL17: What is known and not known. Scandinavian Journal of Immunology 93, e12965 (2021).

[56] S. P. Giblin, J. E. Pease, What defines a chemokine? – The curious case of CXCL17. Cytokine 168, 156224 (2023).

[57] J. Duan, et al., Insights into divalent cation regulation and G13-coupling of orphan receptor GPR35. Cell Discov 8, 1–12 (2022).

4. Can the authors use alpha fold to address whether any of these non-canonical molecules actually is predicted to fold like a chemokine? More generally, based on the paper's analysis, how do the authors propose to define a chemokine? It is well-accepted that chemokines are defined by structure, not function (e.g. limited truncation of any chemokine abrogates activity, but it is still a chemokine structurally, not semantically, folds like a chemokine, aligns with other chemokines).

In response to the recommendation from reviewer 2 to incorporate AlphaFold data, we leveraged AFDB Clusters (foldseek.com), a recently developed tool that clustered over 200 million Uniprot proteins based on their predicted AlphaFold structures (as described in this Nature paper: <https://www.nature.com/articles/s41586-023-06510-w>). We utilised this pre-computed dataset of clustered proteins to query with representative human proteins, both canonical and non-canonical chemokine ligands, and the results are summarised in the table below. Notably, we observed that canonical chemokines

were distributed across different AlphaFold clusters, each corresponding to different ligand types (e.g., CC and CXC). Interestingly, despite this, all these clusters exhibited similar descriptions (e.g. CC or CXC), indicating that the method effectively recovers well-characterized chemokines. Conversely, when analysing non-canonical chemokine ligands, none of them were classified within the canonical chemokine clusters. This observation strongly suggests that canonical and non-canonical ligands do not share the same protein fold. Additionally, we identified intriguing correlations between these structure-based clusters and the results from our phylogenetic analyses. For instance, CXCL14 was clustered within a CC-type group, consistent with our reconciled tree positioning it within the broader CC-type clade (as shown in Figure 2A). Similarly, CXCL16 formed its own unique cluster, which aligns with our CLANS analysis, where it is the last group to connect with canonical chemokines (illustrated in Figure 1A and Figure S1). Furthermore, TAF5 was found in a distinct cluster, mirroring our phylogenetic analyses that place it as the most basal TAF5 clade (as depicted in Figure 2A and Figure S19). While these findings are intriguing, we acknowledge that additional in-depth analyses, beyond the scope of this paper, will be necessary to confirm these results.

In response to the reviewer's inquiry regarding how to define a chemokine, it is essential to recognise that many proteins can exhibit similar 3D structures without being considered homologous. A notable example is the opsins, which are present in both bacteria and animals. Despite sharing a common 3D structure that is characterised by seven transmembrane domains (TMDs) and serves similar functions, they are not regarded as homologous, as highlighted in this study (<https://doi.org/10.1186/gb-2005-6-3-213>). Considering these findings, we propose that, like various other gene families, the primary criterion for assessing protein homology should be rooted in shared evolutionary ancestry and common origin, and this should take precedence over structural similarities.

	Human gene	Uniprot Accession	AFDB Cluster	
			Accession	Description
Canonical CKs	CXCL14	O95715	A0A3Q3M453	C-C motif chemokine
	CCL24	O00175	A0A4X1T574	C-C motif chemokine
	CX3CL1	P78423	A0A7J8CF84	C-X3-C motif chemokine ligand 1
	CXCL1	P09341	A0A1S2ZIJ4	C-X-C motif chemokine
	CXCL13	O43927	A0A1S2ZIJ4	C-X-C motif chemokine
	CXCL8	P10145	A0A1S2ZIJ4	C-X-C motif chemokine
	CCL20	P78556	A0A6P7X7F3	C-X-C motif chemokine
	XCL1	P47992	A0A6P7X7F3	C-X-C motif chemokine
	CXCL16	Q9H2A7	A0A6P8SIS6	C-X-C motif chemokine 16
	CCL27	Q9Y4X3	A0A1L8GBB9	SCY domain-containing protein
	CCL1	P22362	A0A3B4A358	SCY domain-containing protein
	CCL5	P13501	A0A3B4A358	SCY domain-containing protein
	CCL28	Q9NRJ3	A0A3Q0SB19	SCY domain-containing protein
CXCL12	P48061	A0A401SMI2	SCY domain-containing protein	
CXCL17	CXCL17	Q6UXB2	No cluster found	No cluster found
TAF5	TAF5A	Q7Z5A9	Q96LR4	Chemokine-like protein TAF5-4
	TAF5B	Q8N3H0	Q96LR4	Chemokine-like protein TAF5-4
	TAF5C	Q7Z5A8	Q96LR4	Chemokine-like protein TAF5-4
	TAF5D	Q96LR4	Q96LR4	Chemokine-like protein TAF5-4
	TAF5E	Q7Z5A7	A0A7M4EYY1	TAF5 chemokine like family member 5
CYTL	CYTL1	Q9NRR1	A0A673GVE4	Cytokine-like protein 1
CKLF5	CMTM5	Q96DZ9	A0A4W2H069	CKLF like MARVEL transmembrane domain containing 5
	CMTM8	Q8IZV2	U3IR50	CKLF like MARVEL transmembrane domain containing 7

CMTM7	Q96FZ5	A0A6G1PQK5	CKLF-like MARVEL transmembrane domain-containing protein 7
CMTM6	Q9NX76	A0A814ULI9	Hypothetical protein
CKLF	Q9UBR5	A0A3M0K8M7	MARVEL domain-containing protein
CMTM1	Q8IZ96	A0A3M0K8M7	MARVEL domain-containing protein
MAL	P21145	A0A402F5Z5	MARVEL domain-containing protein
CMTM2	Q8TAZ6	A0A6G1S7Y0	MARVEL domain-containing protein
PLP2	Q04941	A0A667IJ27	Proteolipid protein 2
CMTM3	Q96MX0	A0A3B1ILJ1	Zgc:136605
CMTM4	Q8IZR5	A0A3B1ILJ1	Zgc:136605
PLL	Q9Y342	A0A3B1ILJ1	Zgc:136605

5. Chemokine genes are found on many human chromosomes with large clusters on chromosome 2 and 17. Can the authors address the syntenic relationships phylogenetically?

There are cases where synteny data have been used to infer the relationship between species (e.g. <https://doi.org/10.1038/s41586-023-05936-6>); however, to our knowledge, they cannot be used to infer the pattern of gene duplications and losses, as we have done here with gene tree to species tree reconciliations. However, the two approaches are extremely powerful combined and compared as they provide independent evidence. For example, with our phylogenetic analysis of chemokine ligands, we found that CXCL1-10 plus CXCL13 form a monophyletic clade (Figure 2A); this is consistent with their location on the human chromosome 4 (Zlotnik and Yoshie 2012). Similarly, most of the CC-type chemokines, that we find monophyletic in our trees, are located in a locus in human chromosome 17. Likewise, chemokine receptor phylogenetic relationships are largely consistent with macro and micro syntenic patterns. Most of the chemokine receptors are on human chromosome 3 (Zlotnik and Yoshie 2012) and they all belong to a large monophyletic clade in our tree (Figure 4A). Smaller clusters also maintain correspondence, such as the mini cluster of CXCR1 and CXCR2 on human chromosome 2 corresponding to a monophyletic clade in our phylogenetic analysis (Figure 4A).

We have incorporated the above considerations in our manuscript at the lines:

- Lines 140-148 (ligands)
- Lines 256-272 (receptors)
- Lines 375 – 483 (discussion)

6. The authors indicate that 'CXCL8 is present in all jawed vertebrates except in the cartilaginous fishes lineage'. However, they should point out that CXCL8 is not represented in mice. The notion that the repertoire of chemokine and chemokine receptor genes can be different in even closely related species as well as in individuals of the same species is well-documented but not mentioned here.

We thank the reviewer for these suggestions, and we have modified the text in lines 137-138.

7. The analysis suggests that chemokine gene repertoires start small and grow non-linearly to 45 in mammals. However DeVries et al (JI 2005) published that zebrafish have the most chemokines, 63, and chemokine receptors, 24. Do the authors disagree? This should be addressed.

The significant increase in the number of ligands and receptors in zebrafish, compared to their last common mammalian ancestor, can be attributed to an additional round of whole-genome duplication (WGD) ([https://doi.org/10.1016/S0955-0674\(99\)00039-3](https://doi.org/10.1016/S0955-0674(99)00039-3)).

Concerning ligands, the count in zebrafish varies from 63 in DeVries et al. 2005 to 111 in Nomiya et al. 2008, and to 35 in our study. This variation can be attributed to several factors:

1. Genome Versions: The disparities may arise from the use of different versions of the zebrafish genome. We utilised an improved version known for its higher contiguity and reduced fragmentation

(<https://www.nature.com/articles/nature12111>). It is possible that the additional ligands identified by DeVries, Nomiya, and others were partial sequences.

2. Methodology: Methodological differences are at play. DeVries et al. employed tblastN, while we opted for BLASTP. Nomiya et al. do not specify the type of BLAST performed.

3. Stringency: We collected our sequences based on a BLASTP search using as query sequences only manually curated sequences from UniProt. This additional precaution allowed us to identify sequences with high-confidence chemokine ligand characteristics.

4. Sequence Characteristics: Ligands typically have shorter sequences and exhibit less sequence conservation compared to receptors. Zebrafish represents a case in which working with short sequences may lead to missed homologs.

5. Species-Specific Nature: Our approach successfully recovered the complete set of ligands in other species, such as humans and mice. Zebrafish appears to be an exception rather than the norm.

When it comes to receptors, which typically have longer sequences, making it easy to identify distant homologs, our results closely mirror those of DeVries in 2005. In our study, we identified 28 canonical receptors, compared to their count of 24. However, it is worth highlighting that within our dataset, four of these receptors appear as species-specific duplications, potentially indicating that they are actually isoforms or related variants.

Nonetheless, it is essential to emphasise that our work does not aim to precisely reconstruct the entire complement of ligands and receptors in zebrafish or other species. Achieving this would require further validation, including the expression analysis of potential transcripts.

8. Did the authors find any species in which a chemokine/chemokine receptor pair are not found together? That is, if the system is irreducibly complex, requiring both a ligand and receptor, the probability of both genes arising simultaneously is essentially zero. So how do the authors theorize that such a system actually arose, and is there any evidence in their data set for convergence of separately evolved ligand and receptor?

Our data strongly support the hypothesis that the canonical chemokine system originated within the stem group of vertebrates, likely as a consequence of two rounds of genome duplication. This likely accounts for the simultaneous emergence of both ligands and receptors. While the receptors (both canonical and non) can be traced back to a single-gene duplication event (with the exception of ACKR1), the evolution of ligand families capable of interacting with chemokine receptors occurred independently, although further experiments are required to validate this *in vivo* in a broader set of organisms. In our study, we successfully identified the complete set of receptors and ligands in well-established model systems like humans and mice. However, when it comes to interactions between ligands and receptors outside these model organisms, the picture becomes less clear. Similarly, the exact pairings of non-canonical components are also not fully clarified (see lines 404-406). As a result, speculating about evolutionary conservation in these contexts requires caution and further investigation. It's worth noting that chemokines and their corresponding chemokine receptors do not necessarily evolve in tandem. Since they are encoded by different genes, they evolved from separate duplication events occurring at different points in evolutionary history. In certain instances, due to the system's flexibility, chemokines binding orthologous receptors may not be orthologous themselves but may have independently acquired the ability to activate the same receptor in various species.

9. Line 180, 181 and elsewhere: GPCR1 and GPCR33 should be GPR1 and GPR33

We have corrected this throughout the manuscript.

10. Line 185: ACKR1 exceptionalism is noted, but there is no discussion of the remarkable structure-function paradox that the most distantly related chemokine receptor is also the most highly promiscuous receptor, binding many but not all CC and CXC chemokines with high affinity.

We added in the discussion section this consideration regarding the wide binding of ACKR1 (Lines 341-343) and its ability to bind both CC and CXC chemokines ([DOI: 10.1126/science.7689250](https://doi.org/10.1126/science.7689250) and [10.3389/fimmu.2015.00279](https://doi.org/10.3389/fimmu.2015.00279)), highlighting the intriguing contrast with the fact that it is the most distantly related receptor.

11. Line 196: the viral receptors cluster with the vertebrate receptors, suggesting that the viruses captured the receptor

gene from the host. Authors might mention this obvious point regarding origins, and discuss how it relates to the monophyly and paraphyly that emerges from the phylogenetic analysis.

We added a comment to the discussion section (Lines 348-352) regarding the potential origins of the viral chemokine receptors.

12. Any discussion of chemokine-like convergent evolution presupposes that the activity is real and actually occurs *in vivo*. The authors should make clear to what extent the existing literature supports this. As mentioned above, CXCL17 interaction with GPR35 has been challenged *in vitro* and has never been demonstrated to occur *in vivo*. To what extent is the same limitation a problem in considering co-evolution of the other non-canonical chemokines? I agree that classification based solely on function is inappropriate, but so is phylogenetic analysis without direct knowledge of *in vivo* function. It is not feasible to address this in a phylogenetic analysis, but there ought to be at least one species in which the non-canonicals have been rigorously shown to act at specific receptors *in vivo* before grouping them with the canonicals in a co-evolutionary sense.

We agree with the referee that evidence of real chemokine-like activity is important to consider the activity *in vivo*. In our work, the molecules examined were chosen based on previous evidence of chemokine-like sequence similarity, ability to bind canonical components and/or chemokine-like function. For example, CKLF (also called CKLF1) has been shown, through calcium mobilisation and chemotaxis assays using the human cell line HEK293, to bind CCR4 and to induce cell migration via CCR4 respectively (<https://doi.org/10.1016/j.lfs.2005.05.070>). Numerous papers are studying the *in vitro* and *in vivo* effects of CKLF in murein and human models (<https://doi.org/10.1016/j.cyto.2017.12.002>), therefore, we found it compelling to investigate its evolutionary relationship with canonical chemokines. Similarly, CYTL1, that had been predicted to possess an IL8-like fold (<https://doi.org/10.1002/prot.22963>), has been found to bind CCR2 (<https://doi.org/10.4049/jimmunol.1501908>) and *in vitro* and *in vivo* studies showed chemotactic activity for neutrophils (<https://doi.org/10.1007/s10753-019-01116-9>). Ongoing research into this molecule are focusing on a wide array of immune functions (<https://doi.org/10.1007/s00018-019-03137-x>).

We mentioned these considerations in our introduction to explain why we were interested in investigating these molecules (lines 50-57). We have also added a line in the Discussion (lines 323-324) where we reinforce the idea that *in vitro* and *in vivo* experiments for all chemokine-like molecules are required to validate computation predictions.

13. The discussion of homeostatic vs inflammatory chemokine/receptors in the last section of the Discussion would be enhanced by pointing out that the chemokine specificities are numerically totally different for these two groupings, homeostatics tending to have monogamous ligand-receptor relationships and inflammatories being highly promiscuous.

To account for the reviewer's comment, we have added this consideration in a paragraph of the discussion (see Line 389-394).

Reviewer #2 (Significance (Required)):

Much of the paper's results are confirmatory of previous work based on less extensive sequence analysis. One could say more generally that unrelated chemical forms, not just unrelated proteins, have chemokine-like ligand function. For example leukotriene B4 is a powerful leukocyte chemoattractant for neutrophils working through a GPCR. That proteins might also independently evolve common functions does not add insight beyond what is already appreciated. The notion that chemokine receptors have a common ancestor is also generally accepted and that ACKR1 is an outlier is already appreciated. The present work adds phylogenetic and statistical precision to these points.

Our discoveries clarify various aspects of the chemokine system's evolution, and we are confident that the "phylogenetic and statistical precision" of our findings will provide a solid cornerstone for future research aimed at unravelling the function and evolution of the system. Specifically, our work clarified:

1. The presence only in Vertebrates: We have confirmed, through a comprehensive taxonomic sampling (we use many more species than previous works), that the chemokine system is exclusive to vertebrates. However, intriguingly, we identified a TAFA chemokine-like family in urochordates.

2. Relationships between Ligands: We conducted a thorough examination of the relationships between canonical and non-canonical ligands and suggested that several unrelated molecules might have evolved independently their ability to interact with the chemokine receptors. We appreciate the comment of the reviewer regarding the fact that unrelated chemical forms such as leukotriene B4 may have chemokine-like functions. However, in our work all the non-canonical components examined are proteins and as such could have an evolutionary relationship with chemokines. Furthermore, we chose to consider only proteins that showed multiple lines of evidence implicating them in the chemokine system and that are currently the topic of interest in the field (see replies to reviewer 1's comment #5 and to reviewer 2's comment #12). Seeing the general interest in the topic, and especially seeing as this had never been clarified before, in this work, we set ourselves the goal to investigate the evolutionary relationship amongst these non-canonical ligands and canonical chemokines.

3. Duplication Events: We pinpoint the specific gene duplication events responsible for the emergence of chemokine receptors.

4. Atypical Receptor Paraphyly: Our work highlights the paraphyletic nature of atypical receptors, in contrast to previous research (see <https://doi.org/10.1155/2018/9065181>).

5. Viral Receptor Phylogenetics: To our knowledge, this is the first work to investigate the phylogenetic affinities of viral receptors.

6. GPCR182 and Atypical Receptor Affinities: We clarify the affinity of GPCR182 with atypical receptor 3, offering different insights compared to prior studies (see figure S3C in <https://doi.org/10.1038/s41467-020-16664-0>).

7. Additionally, our study represents the first analysis of the chemokine system in the basal vertebrate hagfish and provides insights into the ancestral form of the chemokine system.

8. Ultimately, our research identifies numerous molecules and receptors with potential chemokine functions.

In conclusion, we contribute to resolving uncertainties surrounding the system's origin, including the complex duplication events that have shaped receptor evolution. As evident from the extensive comments provided by the reviewer, our work addresses various controversies in the field (e.g. the inclusion of CXCL17 as a chemokine). Nonetheless, like any new set of findings, our work amalgamates confirmatory results (as highlighted in point 1) with innovative discoveries (as outlined in points 2-8). However, the latter category significantly outweighs the former, underscoring the richness of novel insights. Finally, we would like to thank reviewer 2 for their comments, as these have contributed to greatly improve our manuscript.

December 1, 2023

RE: Life Science Alliance Manuscript #LSA-2023-02471

Dr. Roberto Feuda
University of Leicester
University Road
Leicester LE1 9HN
United Kingdom

Dear Dr. Feuda,

Thank you for submitting your revised manuscript entitled "The origin, evolution and molecular diversity of the chemokine system". We would be happy to publish your paper in Life Science Alliance pending final revisions necessary to meet our formatting guidelines.

- please upload your main and supplementary figures as single files
- please add a Running Title and a Summary Blurb/Alternate Abstract to our system
- please add a Category for your manuscript in our system
- please add the Twitter handle of your host institute/organization as well as your own or/and one of the authors in our system
- please add an Author Contributions section to your main manuscript text and their contributions to the system as well
- please add a conflict of interest statement to your main manuscript text
- please add callouts for Figures S1A-D; S2A-D; S6; S23A to your main manuscript text
- the separate file with supplementary results should be incorporated into the main manuscript, along with those References

A. FINAL FILES:

B. MANUSCRIPT ORGANIZATION AND FORMATTING:

Sincerely,

Reviewer #1 (Comments to the Authors (Required)):

This set of revisions has addressed in detail all my questions.

Reviewer #2 (Comments to the Authors (Required)):

The authors have addressed my comments in their revision and in their reviewer response. I have no further comments. The paper will be a useful, broad and up-to-date review of the chemokines for evolutionary biologists and immunologists.

December 18, 2023

RE: Life Science Alliance Manuscript #LSA-2023-02471R

Dr. Roberto Feuda
University of Leicester
University Road
Leicester LE1 9HN

Dear Dr. Feuda,

Thank you for submitting your Research Article entitled "The origin, evolution and molecular diversity of the chemokine system". It is a pleasure to let you know that your manuscript is now accepted for publication in Life Science Alliance. Congratulations on this interesting work.

DISTRIBUTION OF MATERIALS:

Again, congratulations on a very nice paper. I hope you found the review process to be constructive and are pleased with how the manuscript was handled editorially. We look forward to future exciting submissions from your lab.

Sincerely,
